# Caveolae couple mechanical stress to integrin recycling and activation

**Fidel-Nicolás Lolo[1], Dácil María Pavón[1†], Araceli Grande-García[1‡], Alberto Elosegui-Artola[2§], Valeria Inés Segatori[1#], Sara Sánchez[1], Xavier Trepat[2], Pere Roca-Cusachs[2], Miguel A del Pozo[1]***

[1]Mechanoadaptation and Caveolae Biology Laboratory, Cell and developmental Biology Area, Centro Nacional de Investigaciones Cardiovasculares, Madrid, Spain; [2]Institute for Bioengineering of Catalonia, Barcelona, Spain

**\*For correspondence:**
madelpozo@cnic.es

**Present address:** [†]Allergy Therapeutics S.L., Avda. Punto Es, Alcalá de Henares, Spain; [‡]Structural Biology Programme. Centro Nacional de Investigaciones Oncológicas, Madrid, Spain; [§]Cell and Tissue Mechanobiology Laboratory. The Francis Crick Institute, London, United Kingdom; [#]Center of Molecular and Translational Oncology, Quilmes National University, Bernal, Argentina

**Abstract** Cells are subjected to multiple mechanical inputs throughout their lives. Their ability to detect these environmental cues is called mechanosensing, a process in which integrins play an important role. During cellular mechanosensing, plasma membrane (PM) tension is adjusted to mechanical stress through the buffering action of caveolae; however, little is known about the role of caveolae in early integrin mechanosensing regulation. Here, we show that Cav1KO fibroblasts increase adhesion to FN-coated beads when pulled with magnetic tweezers, as compared to wild type fibroblasts. This phenotype is Rho-independent and mainly derived from increased active β1-integrin content on the surface of Cav1KO fibroblasts. Fluorescence recovery after photobleaching analysis and endocytosis/recycling assays revealed that active β1-integrin is mostly endocytosed through the clathrin independent carrier/glycosylphosphatidyl inositol (GPI)-enriched endocytic compartment pathway and is more rapidly recycled to the PM in Cav1KO fibroblasts, in a Rab4 and PM tension-dependent manner. Moreover, the threshold for PM tension-driven β1-integrin activation is lower in Cav1KO mouse embryonic fibroblasts (MEFs) than in wild type MEFs, through a mechanism dependent on talin activity. Our findings suggest that caveolae couple mechanical stress to integrin cycling and activation, thereby regulating the early steps of the cellular mechanosensing response.

## Editor's evaluation

This valuable cell biological study uses magnetic tweezers to explore how integrins and caveolae interact to regulate mechanosensing. The authors describe a convincing link between the presence of caveolae and the trafficking of integrins between the cell surface and intracellular compartments to control plasma membrane tension.

## Introduction

Cells constantly adjust their plasma membrane (PM) composition in response to changes in extracellular matrix (ECM) stiffness, which regulates many aspects of cell behavior (**Wells, 2008**). How cells sense and react to ECM stiffness changes is critical to understanding both physiological and pathological processes (**Handorf et al., 2015**). The ECM is mechanically linked to the cytoskeleton through integrins, which provide a bridge between extracellular cues and downstream cellular events (**Schwartz, 2010**). The mechanosensing function of integrins is well established, and much progress has been made in defining the molecular details of integrin action; for example, how $\alpha_5\beta_1$ and $\alpha v\beta 3$ integrins withstand and detect forces, respectively (**Roca-Cusachs et al., 2009**), and how differences in integrin bond dynamics contribute to tissue rigidity sensing (**Elosegui-Artola et al., 2014**).

**eLife digest** Cells can physically sense their immediate environment by pulling and pushing through integrins, a type of proteins which connects the inside and outside of a cell by being studded through the cellular membrane. This sensing role can only be performed when integrins are in an active state.

Two main mechanisms regulate the relative amount of active integrins: one controls the activation of the proteins already at the cell surface; the other, known as recycling, impacts how many new integrins are delivered to the membrane. Both processes are affected by changes in cell membrane tension, which is itself controlled by dimples (or 'caveolae' – little caves in Latin) present in the cell surface. Caveolae limit acute changes in tension by taking in (pinching off the dimples) or releasing (dimples flattening) segments of the membrane. However, it is still unclear how integrins and caveolae mechanically interact to regulate the ability for a cell to read its environment.

To understand this process, Lolo et al. focused on mouse cells genetically manipulated to not build caveolae on their surfaces, and which cannot properly sense mechanical changes in their surroundings. These were exposed to beads covered in an integrin-binding protein and manipulated using magnetic tweezers. The manipulation showed that mutated cells bound to the beads more strongly than non-modified cells, indicating that they had more active integrins on their surface. This change was due to both an accelerated recycling mechanism (which resulted in more integrin being brought at the surface) and an increase in integrin activation (which was triggered by a higher membrane tension). Caveolae therefore couple mechanical inputs to integrin recycling and activation.

Healthy tissues rely on cells correctly sensing physical changes in their environment so they can mount an appropriate response. This ability, for example, is altered in cancerous cells which start to form tumours. The findings by Lolo et al. bring together physics and biology to provide new insights into the potential mechanisms causing such impairments.

Integrin function requires an activation step commonly achieved by binding to activator molecules, such as talin and kindlins (*Moser et al., 2009*). Integrins are also activated in response to changes in membrane tension, as recently reported by *Ferraris et al., 2014* and *Böttcher and Fässler, 2014*. However, it is unclear how these events influence mechanosensing.

Tissues experiencing wide variations in PM tension, such as endothelium, muscle, fibroblasts, and adipocytes, have a high-membrane density of caveolae (*Sinha et al., 2011*; *Echarri and Del Pozo, 2015*; *Parton and del Pozo, 2013*). Caveolae are 60–80 nm PM invaginations and are key elements in the sensing and transduction of mechanical forces. Core caveolae protein components are caveolin-1 (Cav1) and cavin-1 (also called Polymerase I and transcript release factor, PTRF), and lack of either results in caveolae loss (*Rothberg et al., 1992*; *Hill et al., 2008*). Cav1 is intimately linked to integrins (*Parton and del Pozo, 2013 del Pozo et al., 2005*; *Del Pozo and Schwartz, 2007*) and to molecules such as Filamin A, which links integrins to the cytoskeleton (*Muriel et al., 2011*). To study how integrins and caveolae interact to regulate cell mechanosensing, we used magnetic tweezers (MT; *Tanase et al., 2007*) to exert mechanical force on mouse embryonic fibroblasts (MEFs) upon their binding to magnetic beads coated with the integrin-binding protein fibronectin (FN). MEFs lacking Cav1 (Cav1KO) showed higher adhesion to FN-coated beads than Cav1WT MEFs, through a process dependent on increased β1-integrin surface availability in Cav1KO MEFs due to both rapid recycling of endocytosed β1-integrin to the PM, in a Rab4 and PM tension-dependent manner, and tension-mediated β1-integrin activation driven by Talin. These results support a role for caveolae in regulating integrin mechanosensing by coupling membrane tension to integrin recycling and activation.

## Results

### Genetic models for studying the role of caveolae in integrin mechanosensing

To study the role of caveolae in integrin mechanosensing, we generated transgenic fibroblast lines by transducing Cav1KO MEFs with either Cav1- or PTRF-expressing retroviral vectors (*Figure 1A–D*).

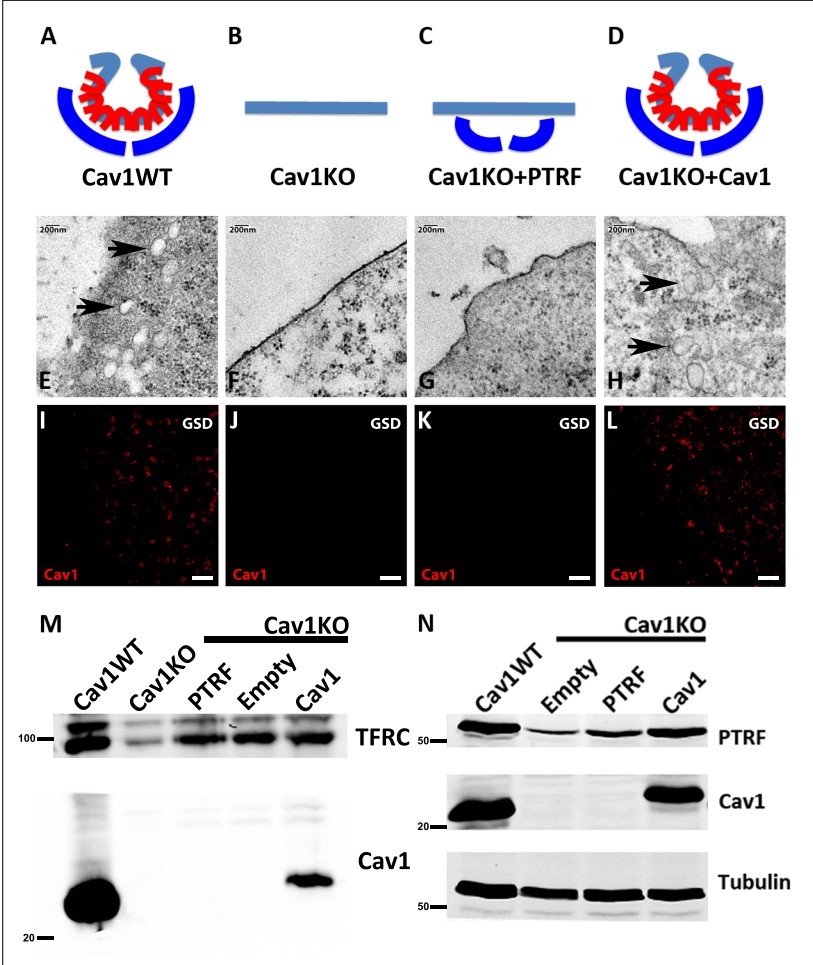

**Figure 1.** Caveolin-1-based genetic model characterization. (**A–D**) Mouse embryonic fibroblast (MEF) caveolae-related phenotypes: PTRF is depicted in dark blue, Cav1 in red, and plasma membrane (PM) in light blue. (**E–H**) Electron microscopy images of MEF PM regions, showing the presence of caveolae only in (**E**) wild type MEFs and (**H**) Cav1-reconstituted Cav1KO MEFs (black arrows). (**I–L**) Ground State Depletion (GSD)-super-resolution MEF images. Cav1 is shown in red. Scale bar = 1 μm. (**M**) Biochemical PM fractionation of wild type and Cav1KO MEFs and Cav1KO MEFs reconstituted with PTRF, empty vector, and Cav1. Samples were immunoblotted for Cav1 and transferrin receptor (as both PM marker and loading control). (**N**) Western blot of total lysates from wild type, Cav1KO, and reconstituted Cav1KO MEFs. Samples were immunoblotted for PTRF, Cav1, and tubulin (loading control).

The online version of this article includes the following source data for figure 1:

**Source data 1.** Full raw unedited blot corresponding to *Figure 1M*.

**Source data 2.** Uncropped blot with the relevant bands labeled, corresponding to *Figure 1M*.

**Source data 3.** Full raw unedited blot corresponding to *Figure 1N*.

**Source data 4.** Uncropped blot with the relevant bands labeled, corresponding to *Figure 1N*.

To validate the transgenic lines for the characterization of mechanosensing properties, we first analyzed the presence or absence of caveolae by electron microscopy (EM), detecting caveolae only in wild type (WT) and Cav1-reconstituted Cav1KO cells (*Figure 1E–H*, black arrows). Next, super-resolution microscopy analysis of Cav1 topographical distribution revealed that re-expressed Cav1 localizes to the PM and does not form aberrant aggregates (*Figure 1I–L*). Confirming the super-resolution imaging data, biochemical fractionation indicated that re-expressed Cav1 localizes to the PM (*Figure 1M*). Western blot analysis confirmed the expected Cav1 and PTRF expression in the different MEF lines and revealed near-endogenous expression levels in Cav1-reconstituted cells (*Figure 1N*, see also *Figure 1—source data 1–4*).

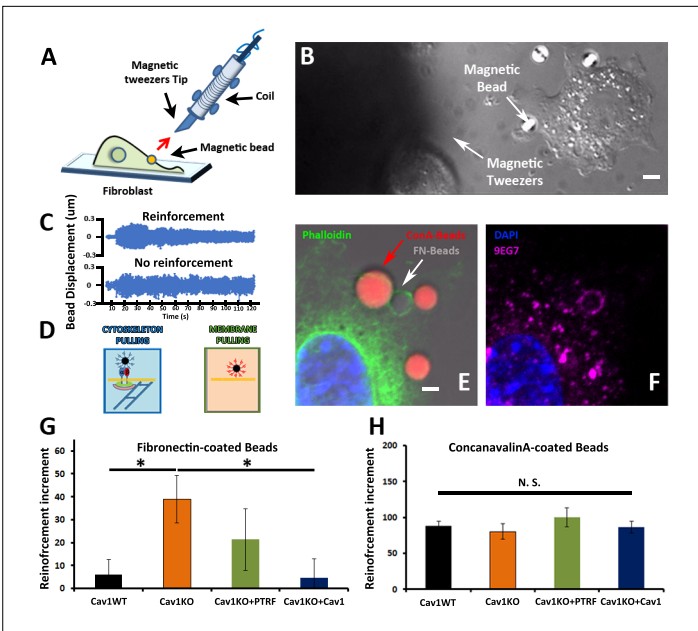

**Figure 2.** Cav1KO Mouse embryonic fibroblasts (MEFs) show reinforced attachment to magnetic tweezers. (**A**) Reinforcement experiment scheme, indicating the fibroblast, the magnetic bead, and the magnetic tweezers apparatus. The red arrow represents the magnetic force exerted on the bead by the magnet. (**B**) Differential interference contrast (DIC) image showing a MEF, the tip of the magnetic tweezers, and a magnetic bead (white arrows). Scale bar = 3 μm. (**C**) Examples of bead oscillation as a function of time in two conditions: with and without reinforcement. (**D**) Representation of the two magnetic beads coatings used: fibronectin (FN), which binds integrins, and ConA, which binds the bulk plasma membrane (PM). (**E and F**) Confocal microscopy images showing a MEF attached to concanavalin A-coated beads (red) and FN-coated beads (gray). Actin staining is shown in green (phalloidin), active β1-integrin in magenta (9EG7 antibody), and 4',6-diamidino-2-phenylindole (DAPI) in blue. Note how only FN-coated beads present both phalloidin and 9EG7 staining. Scale bar = 2 μm. (**G and H**) Reinforcement increment (relative change in reinforcement over the entire experiment, calculated as the difference between the last and initial measurements) of different MEF genotypes for FN-coated beads (**G**) or ConA-coated beads (**H**); n≥20 beads per genotype. Statistical comparisons were by t-test, with significance assigned at *p<0.05. N. S., non-significant. Data are presented as mean values +/- SEM.

The online version of this article includes the following video and source data for figure 2:

**Source data 1.** Traces of beads movement shown in *Figure 2C*.

**Source data 2.** Raw data of experiments from *Figure 2G and 2H*.

**Figure 2—video 1.** Example magnetic tweezers experiment.
https://elifesciences.org/articles/82348/figures#fig2video1

**Figure 2—video 2.** Molecular representation video of the type of forces generated by concanavalin A-coated beads.
https://elifesciences.org/articles/82348/figures#fig2video2

**Figure 2—video 3.** Molecular representation video of the type of forces generated by fibronectin (FN)-coated beads.
https://elifesciences.org/articles/82348/figures#fig2video3

## Cav1 modulates integrin/cytoskeleton-dependent mechanosensing via Rho-independent mechanisms

To monitor the mechanosensing properties of the different transgenic lines, we used the MT technique. In this method, a pulsed magnetic force (1 Hz, 1nN) is applied to pull magnetic beads attached to the cell surface (*Figure 2A and B*, and *Figure 2—video 1*). The magnetic beads oscillate in response to the pulsed force, and the local stiffness of the bead-cell adhesion can be measured as the ratio of the applied force to the bead movement. Cells able to detect the applied force respond through a phenomenon known as reinforcement, by which they progressively strengthen the cell-bead adhesion

site, increasing its stiffness and thus reducing oscillation amplitude. Reinforcement can therefore be quantified as the change in adhesion stiffness over a specified time, providing a measure of cellular mechanosensing (*Figure 2C* and *Figure 2—source data 1* [table]; for more details on the technique, see *Roca-Cusachs et al., 2009*). We studied forces transmitted through integrins (adhesion strength) by coating beads with FN. As a control, we also studied forces transmitted non-specifically through the PM using beads coated with the sugar-binding lectin concanavalin A (ConA; *Figure 2D* and *Figure 2—videos 2 and 3*). We first assessed the tethering specificity of the two coatings by mixing FN-coated beads (prepared with non-labeled BSA) with ConA-coated beads labeled with Alexa 546-conjugated BSA (*Figure 2E and F*). Staining for phalloidin and 9EG7 antibody (which specifically recognizes β1-integrin in its active conformation; *Lenter et al., 1993*) revealed FN-coated beads surrounded by both signals, indicating engagement of both β1-integrin and the cell cytoskeleton; in contrast, ConA-coated beads were excluded from these stainings (*Figure 2E and F*), indicating they are only bound bulk PM, as previously reported (*Gauthier et al., 2011*).

We next analyzed the integrin/cytoskeleton-dependent mechanical response in different MEF lines. Cells were plated for 10 min on FN-coated coverslips. FN- or ConA-coated beads were then allowed to bind to the cell surface and subjected to magnetic pulses. Reinforcement was significantly higher in Cav1KO MEFs exposed to FN-coated beads than in wild type MEFs (*Figure 2G*). Reconstitution of Cav1KO MEFs with recombinant Cav1 rescued the WT phenotype, supporting that the Cav1KO phenotype is specific. Reconstitution of Cav1KO MEFs with PTRF induced a partial recovery that did not reach statistical significance. In contrast, MT pulling upon binding to ConA-coated beads yielded no significant differences across genotypes (*Figure 2H*). These results support an intrinsic role for Cav1/caveolae in determining PM mechanical properties, through integrin-dependent pathways. Integrins link ECM components, notably FN, to the cytoskeleton, transducing external forces into downstream effects and regulating responses such as cell contraction (*Schwartz, 2010*). We previously showed that Cav1 regulates cell contraction by controlling Rho activity through the localization of p190RhoGAP (*Grande-García et al., 2007*) within the PM (*Grande-García et al., 2007*; *Goetz et al., 2011*). In the absence of Cav1, p190RhoGAP localization to liquid order domains increases, where it can bind and inhibit Rho activity (*Goetz et al., 2011*), thus attenuating cell contractility and associated ECM remodeling (*Grande-García et al., 2007*). This reduced cell contractility contrasts with the local stiffening response observed in our MT assays. Because the specific knock down of this p190RhoGAP isoform fully rescues other biomechanical phenotypes in Cav1KO MEFs (*Gauthier et al., 2011*; *Grande-García et al., 2007*), we assessed the impact of stably knocking down p190RhoGAP (*Goetz et al., 2011*) on the increased reinforcement observed in Cav1KO cells (*Figure 3A–C* and *Figure 3—figure supplement 1*, see also *Figure 3—figure supplement 1—source data 1 and 2*). While traction force microscopy confirmed higher overall contractility in p190RhoGAP-depleted Cav1KO MEFs as expected (*Goetz et al., 2011*; *Figure 3D*), MT measurements revealed no decreased reinforcement upon p190RhoGAP knockdown (*Figure 3C*). These observations suggest that Cav1KO MEFs locally respond to FN-coated beads in a Rho-independent manner, as opposed to Rho-dependent whole cell contractility.

## Loss of Cav1 increases FN adhesion and active surface β1-integrin

To explore the role of cell adhesion in Cav1KO MEF reinforcement, we first analyzed the ability of the different MEF lines to adhere to FN- or ConA-coated plates. Whereas adhesion to ConA-coated plates did not differ between genotypes (*Figure 3E*), adhesion to FN was higher in Cav1KO MEFs and PTRF-reconstituted Cav1KO MEFs (*Figure 3F*). Cav1 re-expression rescued the wild type phenotype, supporting a specific role of Cav1 in cellular adhesion. However, PTRF reconstitution did not have a significant impact on Cav1KO MEFs adhesion and was therefore not considered further in subsequent experiments. In additional experiments with different FN concentrations and other integrin-dependent coatings (collagen and vitronectin), Cav1KO MEFs showed always higher adhesion than wild type MEFs (*Figure 3—figure supplement 1B, C and D*). The two major FN receptors are integrins $\alpha_5\beta_1$ and $\alpha_v\beta_3$ (*Plow et al., 2000*); however, adhesion strength is mainly mediated by the clustering and activation (*Friedland et al., 2009*; *Lin et al., 2013*) of integrin $\alpha_5\beta_1$. To evaluate the role of β1-integrin in reinforcement, we imaged the active β1-integrin pool in Cav1KO and wild type MEFs by staining with 9EG7. The active β1-integrin signal was consistently stronger in Cav1KO MEFs than in wild type MEFs over a range of conditions: permeabilized and non-permeabilized cells, around FN-coated beads and

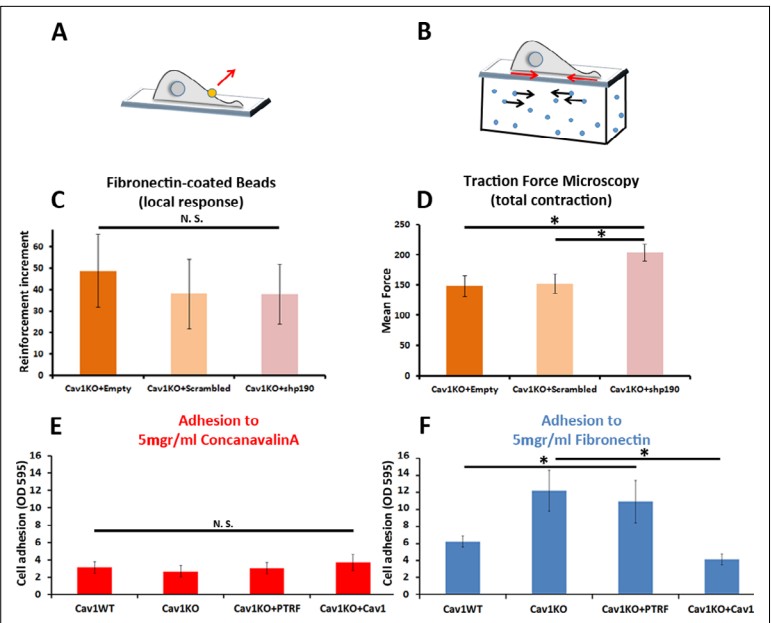

**Figure 3.** Cav1KO Mouse embryonic fibroblasts (MEFs) show Rho-independent reinforcement and increased fibronectin (FN) adhesion. (**A and B**) Experimental schemes for (**A**) local force measurement (magnetic tweezers experiment, reinforcement) and (**B**) total force measurement (traction force microscopy, total cell contraction). (**C**) Effect of transfection with scrambled or shp190RhoGAP small hairpin RNA (shRNA) on the reinforcement increment in Cav1KO MEFs (magnetic tweezers assay). Reinforcement increment refers to the relative change in reinforcement over the entire experiment, calculated as the difference between the last and initial measurements; n≥20 beads per condition. (**D**) Effect of transfection with scrambled or shp190RhoGAP shRNA on mean total force contraction in Cav1KO MEFs (traction force microscopy); n≥12 cells per condition. (**E and F**) Relative adhesion of the indicated genotypes to plates coated with (**E**) 5 µg/ml ConA or (**F**) 5 µg/ml of FN. Measurements (absorbance, optical density, OD, at 595 nm from retained crystal violet dye, see Materials and methods) were normalized to values from adhesion to BSA-coated plates; n≥9 adhesion independent experiments. Data are presented as mean values +/- SEM. Statistical comparisons were by t-test, with significance assigned at *p<0.05. N. S., non-significant.

The online version of this article includes the following source data and figure supplement(s) for figure 3:

**Source data 1.** Raw data of experiments from *Figures 3C–F*.

**Figure supplement 1.** Cav1KO Mouse embryonic fibroblasts (MEFs) show increased adhesion to different substrates and increased active β1 integrin.

**Figure supplement 1—source data 1.** Full raw unedited blot corresponding to *Figure 3—figure supplement 1A*.

**Figure supplement 1—source data 2.** Uncropped blot with the relevant bands labeled, corresponding to *Figure 3—figure supplement 1A*.

**Figure supplement 1—source data 3.** Raw data of experiments from *Figure 3–figure supplement 1*.

at different spreading time points (*Figure 3—figure supplement 1E–M*). Cav1KO MEFs displayed stronger FN adhesion than wild type MEFs, a phenotype presumably derived from increased active β1-integrin at the cell surface.

## Cav1KO MEFs recycle β1-integrin faster than wild type MEFs

We first hypothesized that enhanced recruitment of active β1-integrin to FN beads in Cav1KO MEFs could be derived from increased lateral mobility. However, experiments measuring fluorescence recovery after photobleaching (FRAP) in Cav1KO and wild type MEFs expressing similar levels of GFP-fused β1-integrin (*Figure 4A, B*) revealed no significant differences in fluorescence recovery between the two cell populations, indicating that lateral mobility of β1-integrin was unaffected (*Figure 4C*). Total Internal Reflection Fluorescence (TIRF) videos revealed β1-integrins in wild type MEFs as stable, largely immobile structures; in contrast, β1-integrins in Cav1KO MEFs showed a dynamic behavior, rapidly appearing and disappearing from the PM plane (*Figure 4—videos 1 and 2* and *Figure 4—figure*

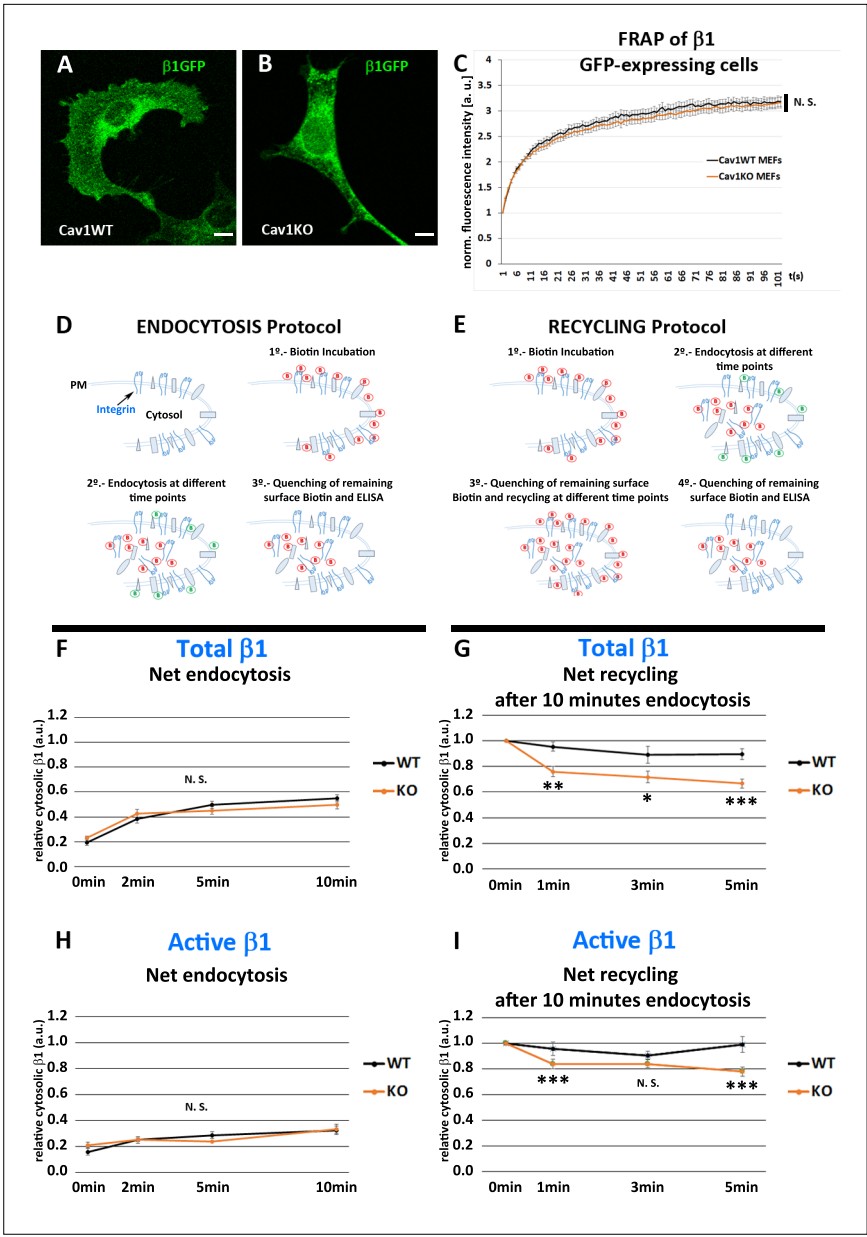

**Figure 4.** Cav1KO mouse embryonic fibroblasts (MEFs) show faster β1-integrin recycling. (**A and B**) Confocal microscopy images of wild type and Cav1KO MEFs transfected with β1-integrin-GFP expression vector. Scale bar = 10 μm. (**C**) Normalized fluorescence intensity recovery after photo bleaching of wild type MEFs (black line) and Cav1KO MEFs (red line) at the indicated time points (the graph is representative of a minimum of nine independent experiments in which between 8 and 10 cells were bleached per genotype). Statistical significance of between time point differences was estimated by t-test; N. S., non-significant. (**D and E**) Experimental schemes for the analysis of endocytosis and recycling followed by ELISA (according to *Li et al., 2016*). (**F and H**) Net endocytosis of (**F**) total β1 integrin and (**H**) active β1-integrin in wild type and Cav1KO MEFs at the time points indicated. Net endocytosis is expressed as internalized biotinylated β1 integrin (cytosolic) at each time point normalized to total biotinylated β1-integrin (internal and surface bound; see Materials and methods); n≥6 endocytosis assays per genotype. (**G and I**) Net recycling after 10 min endocytosis of (**G**) total and (**I**) active β1-integrin in wild type and Cav1KO MEFs at the time points indicated. Net recycling is expressed as internal biotinylated β1 integrin (cytosolic) at each time point normalized to time point 0 (which contains all the biotinylated β1 integrin internalized after 10 min of endocytosis, see Materials and methods); then, decreasing values mean increased recycling; n=10 recycling assays per genotype. Data are presented as mean values +/- SEM. Statistical comparisons were by t-test, with significance of between-group differences denoted *p<0.05, **p<0.01, or ***p<0.001. N. S., non-significant.

*Figure 4 continued on next page*

*Figure 4 continued*

The online version of this article includes the following video, source data, and figure supplement(s) for figure 4:

**Source data 1.** Raw data of experiments from *Figure 4.*

**Figure supplement 1.** Cav1KO mouse embryonic fibroblasts (MEFs) show faster β1-integrin dynamics.

**Figure supplement 1—source data 1.** Raw data of experiments from *Figure 4-figure supplement 1*.

**Figure 4—video 1.** TIRFm video of a Cav1WT mouse embryonic fibroblast (MEF) transfected with β1-gfp expression vector.

https://elifesciences.org/articles/82348/figures#fig4video1

**Figure 4—video 2.** TIRFm video of a Cav1KO mouse embryonic fibroblast (MEF) transfected with β1-gfp expression vector.

https://elifesciences.org/articles/82348/figures#fig4video2

**Figure 4—video 3.** TIRFm video of a Cav1WT mouse embryonic fibroblast (MEF) transfected with β1-gfp expression vector, before hypoosmotic treatment.

https://elifesciences.org/articles/82348/figures#fig4video3

**Figure 4—video 4.** TIRFm video of another Cav1WT mouse embryonic fibroblast (MEF) transfected with β1-gfp expression vector, before hypoosmotic treatment.

https://elifesciences.org/articles/82348/figures#fig4video4

**Figure 4—video 5.** TIRFm video of Cav1WT mouse embryonic fibroblast (MEF) transfected with β1-gfp expression vector from Figure 4-video 3, during hypoosmotic treatment (1:20 DMEM dilution).

https://elifesciences.org/articles/82348/figures#fig4video5

**Figure 4—video 6.** TIRFm video of a Cav1WT mouse embryonic fibroblast (MEF) transfected with β1-gfp expression vector from Figure 4-video 4, during hypoosmotic treatment (1:20 DMEM dilution).

https://elifesciences.org/articles/82348/figures#fig4video6

supplement 1A and 1B). Interestingly, integrins increased their dynamicity after treatment with high hypoosmotic pressure in wild type MEFs, mimicking Cav1KO phenotype (*Figure 4—figure supplement 1C–F*, and *Figure 4—videos 3–6*, quantified in *Figure 4—figure supplement 1F*, see Materials and methods for more details), suggesting that PM tension changes could affect integrin trafficking dynamics. In adherent cells, integrins undergo constant endocytic-exocytic shuttling to facilitate the dynamic regulation of cell adhesion (*Bretscher and Aguado-Velasco, 1998*). To study the effect of these dynamics on β1-integrin surface availability across our tested genotypes, we performed a series of endocytosis/recycling assays with an ELISA-based protocol (*Li et al., 2016*; *Roberts et al., 2001*; *Figure 4D and E*). We first analyzed the endocytic rates of total and active β1-integrin at early time points (2, 5, and 10 min) during early spreading (2 hr after seeding), to recapitulate the conditions used in MT experiments (see Materials and methods for details). Cav1KO and wild type MEFs showed no significant differences, indicating that β1 endocytosis is Cav1-independent at these early time points (*Figure 4F and H*). We next studied the recycling rates after allowing β1 endocytosis to proceed for 5 or 10 min; recycling was tested over two time point sets: 2, 5, and 10 min and 1, 3, and 5 min. After 5 min of endocytosis, wild type and Cav1KO MEFs showed no major differences in total β1-integrin recycling rates at either time point set (*Figure 4—figure supplement 1G, H*). In contrast, after 10 min endocytosis, Cav1KO MEFs recycled total and active β1-integrin faster than wild type MEFs in the 1-3-5 min recycling set (*Figure 4G and I*). Importantly, reconstitution of Cav1KO MEFs with recombinant Cav1 rescued active β1-integrin recycling but not total β1-integrin, supporting that the Cav1KO phenotype is specific at least for the active pool (*Figure 4—figure supplement 1I, J*). These results suggest that β1-integrin is stabilized in the presence of Cav1 after 10 min endocytosis, whereas in Cav1KO MEFs its shuttling to PM is accelerated, increasing its surface availability and thus enhancing adhesion to FN-coated beads and reinforcement response. Different mechanisms could account for this stabilization; however, recent work in our lab showed increased exocytosis after loading wild type MEFs with cholesterol, phenocopying Cav1KO MEFs where cholesterol accrued in different endosomal compartments (*Albacete-Albacete et al., 2020*). To test if cholesterol could play a role in β1-integrin stabilization, we treated wild MEFs with either U18666A (which promotes cholesterol accumulation in endosomal compartments *Cenedella, 2009*) or low-density lipoproteins (LDLs) (which also increases cholesterol content) and analyzed β1-integrin levels by ELISA. Interestingly, both treatments induced a significant increase in surface active β1-integrin (*Figure 4—figure*

*supplement 1K*) that was also accompanied by an increased colocalization with EEA-1 positive endosomes in LDL-treated cells (*Figure 4—figure supplement 1L–N*). These results might be indicative of a Cav1-dependent cholesterol threshold above which β1-integrin trafficking is altered, phenocopying Cav1KO MEFs where both surface and intracellular active β1-integrin levels are increased.

## Clathrin independent carrier/GPI enriched endocytic compartment-dependent uptake contributes to β1-integrin endocytosis in Cav1KO MEFs

β1-integrin endocytosis has been suggested to be Cav1-dependent in different systems (*Shi and Sottile, 2008*; *Du et al., 2011*); however, we observed no differences between Cav1KO and wild type MEFs in net β1-integrin endocytosis. This discrepancy might reflect assay timings, as we studied early time points (2, 5, and 10 min), whereas previous studies focused on endocytosis over longer time periods. To investigate how Cav1KO MEFs achieve similar levels of β1-integrin endocytosis as wild type MEFs, we decided to analyze endocytosis via clathrin independent carriers (CLICs), especially because this endocytic modality is negatively regulated by caveolar proteins, including Cav1 and PTRF (*Chaudhary et al., 2014*; *Cheng et al., 2010*). Furthermore, this endocytic route is highly relevant for fibroblast trafficking dynamics—accounting for ~threefold internalized volume as compared to clathrin-mediated endocytosis—and β1-integrin is a specific cargo (*Howes et al., 2010*;

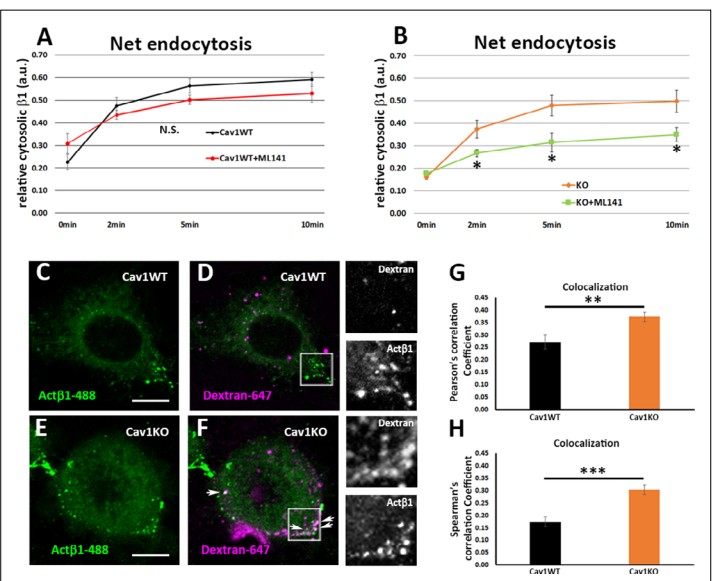

**Figure 5.** Cav1KO mouse embryonic fibroblasts (MEFs) take up β1-integrin by clathrin independent carrier (CLIC) endocytosis. (**A and B**) Net endocytosis (normalized to Total biotinylated β1-integrin) at the time points indicated. (**A**) wild type MEFs treated with ML141 (red line) and untreated controls (black line). (**B**) Cav1KO MEFs treated with ML141 (green line) and untreated controls (orange line); n≥5 endocytosis assays per genotype. (**C–F**) Confocal microscopy images of wild type MEFs (**C and D**) and Cav1KO MEFs (**E and F**) incubated with anti-active β1-Alexa 488 antibody (green) for 1 hr at 4°C followed by incubation with dextran-Alexa 647 (magenta) for 3 min at 37°C. White arrows in F mark colocalization between β1–488–positive particles and dextran-647–positive particles in Cav1KO MEFs. Insets of both Cav1WT and Cav1KO (right side of each panel) show colocalizing structures. Scale bar = 10 µm. (**G and H**) Quantification of colocalization between active β1-Alexa 488 and dextran-647, expressed as Pearson's correlation coefficient (**G**) or Spearman's correlation coefficient normalized by the mean of Cav1WT MEFs (**H**); n≥18 cells per genotype. Data are presented as mean values +/- SEM. Statistical significance of differences across indicated conditions was assessed by t-test: *p<0.05 P**p<0.01; ***p<0.001 N. S., non-significant.

The online version of this article includes the following source data and figure supplement(s) for figure 5:

**Source data 1.** Raw data of experiments from *Figure 5*.

**Figure supplement 1.** Cav1KO mouse embryonic fibroblasts (MEFs) show increased CLIC-dependent β1 integrin endocytosis.

**Figure supplement 1—source data 1.** Raw data of experiments from *Figure 5—figure supplement 1*.

*Lakshminarayan et al., 2014*). Interestingly, cells lacking Cav1 increase CLIC endocytosis through the small GTPase Cdc42 activation (*Cheng et al., 2010*), which is known to regulate this endocytic pathway (*Mayor et al., 2014*). We therefore asked ourselves whether Cav1KO MEFs might preferentially endocytose β1-integrin via this mechanism. We treated wild type and Cav1KO MEFs with the Cdc42 inhibitor ML141 (*Surviladze, 2010*), which inhibits the CLIC-GEEC (GPI enriched endocytic compartment) pathway of endocytosis, involved in fluid-phase uptake and the entry of many specific cargoes, including integrins (*Howes et al., 2010*; *Thottacherry et al., 2017*). ML141 significantly reduced β1-integrin endocytosis in Cav1KO MEFs, whereas wild type MEFs were unaffected (*Figure 5A and B*), indicating that β1-integrin is partially endocytosed through the CLIC/GEEC pathway in Cav1KO MEFs. In accordance with this finding, Cav1KO MEFs displayed significantly higher uptake of the fluid-phase endocytosis marker dextran (*Sabharanjak et al., 2002*), as well as elevated colocalization of dextran and Alexa-488-labeled β1-integrin (*Figure 5C–H* and *Figure 5—figure supplement 1*), as compared to wild type cells. Thus, in the absence of Cav1, early endocytosis of β1-integrin occurs at least in part through CLIC uptake, which provides an alternative entry route that would compensate for lack of Cav1-dependent internalization. To further delineate the relative contribution of the different endocytic routes to β1 integrin endocytosis, we performed a series of colocalization studies of active β1-integrin and previously characterized markers for caveolar-dependent (BODIPY-LacCer), CLIC-dependent (CD44) and clathrin-dependent (transferrin, Tnf-568) endocytosis (; *Cheng et al., 2006*; *Singh et al., 2003*; *Harding et al., 1983*). Consistent with previous results, β1-integrin endocytosis was mainly Cav1-dependent in wild type MEFs as it: (i) colocalized with Cav1 and LacCer, (ii) was significantly reduced upon genistein treatment (a caveolar endocytosis inhibitor *Rejman et al., 2005*), and (iii) was unaffected by ML141 treatment (the CLIC inhibitor; *Figure 5—figure supplement 1D–K*). On the other hand, β1-integrin endocytosis was mainly CLIC-dependent in Cav1KO MEFs as it: (i) colocalized with CD44 and (ii) was significantly reduced upon ML141 treatment as compared to wild type MEFs (*Figure 5—figure supplement 1L–P*). Finally, no significant differences were found in clathrin-dependent β1 integrin endocytosis between wild type and Cav1KO MEFs (*Figure 5—figure supplement 1Q–S*). Altogether, these results further prove that β1-integrin endocytosis is mainly endocytosed by CLIC-dependent mechanisms in Cav1KO MEFs.

## Cav1 is required for Rab11-dependent recycling of β1-integrin

β1-integrin follows the canonical Rab21-Rab11-dependent endosomal trafficking route—which takes longer times to recycle back to the PM—in wild type MEFs; while other paralogs, such as, for instance, β3-integrin follow a Rab4-dependent 'short' loop (*Roberts et al., 2004*; *Roberts et al., 2001*). As stated above, this scenario is in accordance with integrin β1 being localized to Rab11-positive endosomal compartments after 10 min endocytosis in wild type MEFs. In contrast, in the absence of Cav1, β1-integrin is partially sorted to a CLIC-dependent endosomal compartment in Cav1KO MEFs, from which it might be recycled to the PM following different dynamics. We have previously reported that integrins are rapidly delivered to nascent focal contacts in the absence of Cav1 (*Grande-García et al., 2007*). The recycling of CLIC cargo proteins is controlled by a number of factors including several Rabs such as Rab22a (*Weigert et al., 2004*), which has been shown to collaborate with the microtubule and tethering protein HOOK1 during this process (*Maldonado-Báez et al., 2013*). Interestingly, knocking down HOOK1 shifts the trafficking of CLIC cargo proteins from recycling to endosomal targeting, such as the surface glicoprotein CD147, which accumulates in the early endosomal antigen-1 positive compartment (EEA-1) (*Maldonado-Báez et al., 2013*). To analyze whether HOOK1 is required for active β1–integrin recycling, we transfected Cav1KO MEFs with siRNA against HOOK1 and studied the colocalization of EEA1 and 9EG7 (i.e. active β1–integrin) immunolabeling. Whereas HOOK1-deficient Cav1KO MEFs showed increased CD147 and EEA1 colocalization as expected (*Maldonado-Báez et al., 2013*), no significant differences were observed for β1–integrin (*Figure 6A-E*). Accordingly, surface active β1–integrin showed similar levels in both HOOK1-deficient and control Cav1KO MEFs, even 72 hr after siRNA treatment (*Figure 6—figure supplement 1A*). These results indicate that HOOK1 is not required for β1–integrin recycling in Cav1KO MEFs. We then decided to analyze the contribution of canonical Rab11 and Rab4-dependent recycling routes, using dominant-negative (DN) mutants described previously: Rab11 N124I, and Rab4 S22N, respectively (*Roberts et al., 2001*; *White et al., 2007*). We first assessed the impact of disrupting 'long loop'-dependent recycling upon expression of the Rab11 DN mutant (*White et al., 2007*), as assessed by the degree of colocalization

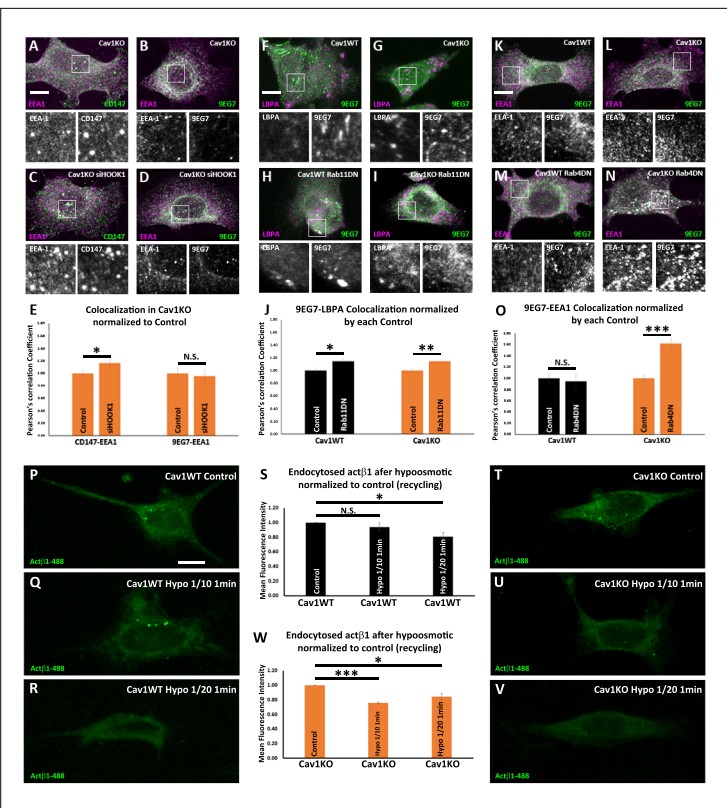

**Figure 6.** Cav1 is required for β1-integrin Rab11-dependent recycling. (**A–D**) Confocal microscopy images of Cav1KO mouse embryonic fibroblasts (MEFs) stained for CD147 (green in A and C), active β1-integrin (9EG7 antibody, green in B and D) and EEA1 (magenta), either non-treated (**A and B**) or treated with siRNA against HOOK1 for 48 hr (**C and D**). Insets (below each panel) show colocalizing structures. Scale bar = 10 µm. (**E**) Quantification of colocalization between EEA1 and CD147 or 9EG7 normalized to control, expressed as Pearson's correlation coefficient normalized by the mean of the corresponding control condition as indicated; n≥20 cells per condition. (**F–I**) Confocal microscopy images of wild type (**F and H**) or Cav1KO MEFs (**G and I**), stained for active β1-integrin (9EG7 antibody, green) and lysobisphosphatidic acid (LBPA) (magenta), either non-transfected (**F and G**) or transfected with Rab11 N124I dominant negative mutant for 48 hr (**H and I**). Insets (below each panel) show colocalizing structures. Scale bar = 10 µm. (**J**) Quantification of colocalization between LBPA and 9EG7 normalized by each control, expressed as Pearson's correlation coefficient normalized by the mean of the corresponding control condition as indicated; n≥20 cells per condition. (**K–N**) Confocal microscopy images of wild type (**K and M**) or Cav1KO MEFs (**L and N**), stained for active β1-integrin (9EG7 antibody, green) and EEA-1 (magenta), either non-transfected (**K and L**) or transfected with Rab4 S22N dominant negative mutant for 48 hr (**M and N**). Insets (below each panel) show colocalizing structures. Scale bar = 10 µm. (**O**) Quantification of colocalization between EEA-1 and 9EG7 normalized to each control, expressed as Pearson's correlation coefficient normalized by the mean of the corresponding control condition as indicated; n≥30 cells per condition. Colocalization was analyzed using the plugin intensity correlation analysis (Fiji *Li et al., 2004*). (**P–V**) Confocal microscopy images of wild type MEFs (**P–R**) and Cav1KO MEFs (**T–V**) incubated with anti-active β1-Alexa 488 antibody (green) for 1 hr at 4°C followed by 3 min endocytosis at 37°C. Remaining surface fluorescence was removed by acid stripping prior to fixation with paraformaldehyde (PFA) at 4%. Scale bar = 10 µm. Quantification of mean fluorescence intensity of endocytosed active β1-integrin in Cav1WT (**S**) or Cav1KO MEFs (**W**). Values were normalized to mean fluorescence intensity of control; n≥30 cells per genotype and condition. Data are presented as mean values +/- SEM. Statistical significance of differences across indicated conditions was assessed by t-test: * p<0.05; P**p<0.01; ***p<0.001 N. S., non-significant.

The online version of this article includes the following source data and figure supplement(s) for figure 6:

**Source data 1.** Raw data of experiments from *Figure 6*.

**Figure supplement 1.** β1 integrin surface levels.

**Figure supplement 1—source data 1.** Raw data of experiments from *Figure 6-figure supplement 1*.

between the endosomal compartment (LBPA, lysobisphosphatidic acid, a late endosomal marker) and active β1–integrin (9EG7 label) in either wild type or Cav1KO cells. While we observed significant differences in the colocalization of 9EG7 and LBPA labels when comparing Cav1KO cells transfected with the Rab11 DN mutant to non-transfected Cav1KO cells, no significant differences were observed on the surface exposure of active β1–integrin in the same cells (*Figure 6G, I and J* and *Figure 6— figure supplement 1B*). In contrast, wild type MEFs showed increased colocalization between 9EG7 and LBPA-positive vesicles (*Figure 6F, H and J*) and reduced surface active β1–integrin levels upon Rab11 DN transfection (*Figure 6—figure supplement 1B*), suggesting that this is the main recycling pathway in wild type cells. We then assessed the impact of expressing a Rab4 S22N DN mutant (which blocks a 'short loop'-dependent recycling *Roberts et al., 2001*) on the trafficking of active β1–integrin in either wild type or Cav1KO cells. Whereas no significant differences in 9EG7-EEA1 colocalization was observed upon disrupting Rab4-dependent trafficking in wild type MEFs, expression of the Rab4 DN mutant increased the colocalization between both labels in Cav1KO cells (*Figure 6K–O*). This was consistent with a significant reduction in surface active β1–integrin levels in Cav1KO cells expressing the Rab4 DN mutant, as compared to non-transfected Cav1KO cells, while no difference was observed for wild type cells (*Figure 6—figure supplement 1C*). Taken together, these results suggest that in the absence of Cav1, β1–integrin recycling is partially switched from 'slow' Rab11-dependent to 'fast' Rab4-dependent, recycling.

## Hypoosmotic shock increases β1-integrin recycling and activation in Cav1KO MEFs

Many studies have shown that membrane trafficking and membrane tension are tightly coupled (*Gauthier et al., 2012*; *Apodaca, 2002*). For example, increases in membrane tension increase exocytosis from the endocytic recycling compartment (*Gauthier et al., 2009*). Cav1KO MEFs lacking caveolae cannot buffer membrane tension properly (*Sinha et al., 2011*). This can lead to increased exocytosis (*Osmani et al., 2018*), as we observed in the specific case of β1-integrin recycling, which in turn increases adhesion to FN-coated beads. To specifically link integrin recycling to caveolae buffering in response to mechanical stress, we incubated Cav1KO and Cav1WT MEFs with anti-active beta1-Alexa 488 antibody for 1 hr at 4°C followed by 3 min endocytosis at 37°C. Immediately after that, we challenged cells for 1 min with either DMEM diluted 1:10 (which can be buffered by caveolae flattening, as previously described *Sinha et al., 2011*, and is shown in our own results, see below in both *Figure 7A-F* and *Figure 7—figure supplement 1N-U*), or DMEM diluted 1:20 in distilled water, which exceeds the buffering capacity of caveolae (as we show below in *Figure 7—figure supplement 1N–U*). To remove any remaining surface fluorescence, two quick steps of acid stripping were done prior to fixation. Strikingly, whereas Cav1KO MEFs showed a significant reduction in the endocytosed active beta1-integrin pool both at 1:10 and 1:20 dilutions (*Figure 6T–W*), Cav1WT MEFs only showed a similar significant reduction at 1:20 dilution (*Figure 6P–S*). These results indicate that caveolae buffering prevents integrin recycling below a certain force threshold and therefore regulate integrin dynamics in response to mechanical stress. However, PM tension can also affect adhesion more directly through increasing integrin activation independently from the cytoskeleton (*Wang et al., 2015*) and in a ligand-independent mechanism (*Ferraris et al., 2014*). To study the role of caveolae in membrane-tension-induced β1-integrin activation, we exposed cells to hypoosmotic conditions. Cav1KO and wild type MEFs were incubated for 10 min in DMEM diluted 1:10 in distilled water, fixed, and stained for 9EG7. Hypoosmotic shock sharply increased β1-integrin activation in Cav1KO MEFs, whereas no significant change was observed in wild type MEFs (*Figure 7A–F*; similar results were obtained with hypoosmotic shock exposure for 30 s and 1 min; data not shown). We also studied the amount of active β1-integrin around FN-coated beads before and after magnetic twisting (*Figure 7—video 1*), observing a significant tension-induced increase in Cav1KO MEFs (*Figure 7—figure supplement 1A–J*). We confirmed that this phenotype is caveolae-dependent, Cav1-independent, as PTRFKO MEFs (that lack caveolae but still express Cav1 *Hill et al., 2008*) also show increased β1-integrin activation upon hypoosmotic shock (*Figure 7—figure supplement 1K–M*). Strikingly, β1-integrin activation was also observed in Cav1WT MEFs after both longer hypoosmotic treatments and with higher hypoosmotic pressures (*Figure 7—figure supplement 1N–W*). To rule out any possible antibody penetration due to paraformaldehyde (PFA)-induced PM disruption, we confirmed these results by incubating MEFs with 9EG7 at 4°C followed by the different hypoosmotic treatments (*Figure 7—figure supplement*

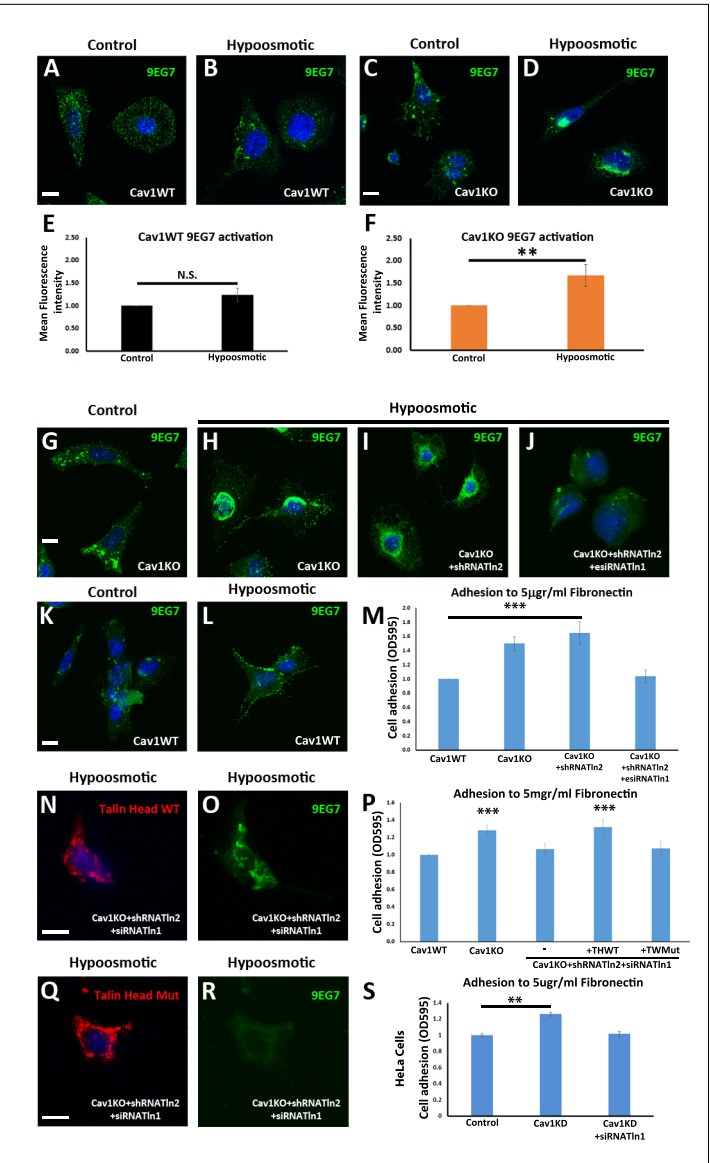

**Figure 7.** Talin is required for the enhanced adhesion and β1-integrin activation phenotype of Cav1KO mouse embryonic fibroblasts (MEFs). (**A–D**) Confocal microscopy images of wild type MEFs (**A and B**) and Cav1KO MEFs (**C and D**) stained for active β1-integrin (9EG7 antibody, green) after culture in standard medium (control in A and C) or hypoosmotic medium (diluted 1:10; **B and D**) for 10 min at 37°C. DAPI is shown in blue. Scale bar = 10 µm. (**E and F**) Quantification of 9EG7 mean fluorescence intensity in control and hypoosmotic shock-exposed wild type (**E**) or Cav1KO MEFs (**F**). Values were normalized by each analyzed area and finally referred to area control; n≥35 cells per genotype. (**G–J**) Confocal microscopy images of active β1-integrin staining (9EG7 antibody, green) in Cav1KO MEFs subject to indicated RNAi treatments and cultured for 10 min at 37°C in standard medium (control) or hypoosmotic medium (diluted 1:10). Scale bar = 10 µm. (**K and L**) Active β1-integrin immunostaining in wild type MEFs cultured in standard (control) or hypoosmotic medium. DAPI counterstain is shown in blue. Scale bar = 10 µm. (**M, P, and S**) Relative adhesion of MEFs of the indicated genotypes (**M and P**) or HeLa Cells (S, wild type – control -; knocking down Cav1 -Cav1KD-; and knocking down both Cav1 and Talin1 -Cav1KD + siTnl1) to plates coated with 5 µg/ml fibronectin (FN). Values were normalized to control condition; n≥18 cells in three independent adhesion experiments. THWT: Talin head wild type; THMut: Talin head mutant. (**N, O, Q, and R**) Confocal microscopy images of talin head domain (**N and Q**) and active β1-integrin (9EG7 antibody; **O and R**) in Cav1KO MEFs transfected with Tln2 shRNA and Tln1 siRNA plus either WT Talin head (**N and O**) or mutant Talin head (**Q and R**). Before immunostaining, cells were cultured for 10 min at 37°C in hypoosmotic medium (diluted 1:10). DAPI is shown in blue. Scale bar = 10 µm. All immunostainings in this figure were performed following the extracellular staining after fixation (see Materials and methods for more details). Data are presented as mean values +/- SEM.

*Figure 7 continued on next page*

*Figure 7 continued*

Statistical significance of differences across indicated conditions was assessed by t-test: * p<0.05 P**p<0.01; ***p<0.001 N. S., non-significant.

The online version of this article includes the following video, source data, and figure supplement(s) for figure 7:

**Source data 1.** Raw data of experiments from *Figure 7.*

**Figure supplement 1.** Cav1KO mouse embryonic fibroblasts (MEFs) show a lower β1 integrin activation threshold in response to mechanical stress.

**Figure supplement 1—source data 1.** Raw data of experiments from *Figure 7-figure supplement 1I, J, M, V and W*.

**Figure supplement 2.** Talin regulates adhesion and β1-integrin activation in Cav1KO MEFs.

**Figure supplement 2—source data 1.** Raw data of experiments from *Figure 7-figure supplement 2*.

**Figure supplement 2—source data 2.** Full raw unedited blot (corresponding to *Figure 7—figure supplement 2K*).

**Figure supplement 2—source data 3.** Uncropped blot with the relevant bands labelled (corresponding to *Figure 7—figure supplement 2K*).

**Figure 7—video 1.** Example magnetic twisting experiment.
https://elifesciences.org/articles/82348/figures#fig7video1

*2A–J*). Altogether, these results might indicate that caveolae restrict integrin activation upon changes in PM tension until their buffering capacity is exhausted; consequently, Cav1KO MEFs lack the ability to adapt integrin activation to mechanical stress.

## Talin supports increased β1-integrin activation and recycling in Cav1KO MEFs

Integrin activation is controlled by a number of intracellular proteins (*Moser et al., 2009*). One of them, talin, regulates integrin adhesion strength by interacting through its amino-terminal FERM domain (the talin head domain; *Tanentzapf and Brown, 2006*). This interaction is necessary and sufficient to induce inside-out integrin activation (*Tadokoro et al., 2003*; *Zhang et al., 2008*). To study the possible role of talin in β1-integrin activation in Cav1KO MEFs, we first confirmed β1-integrin binding competence after hypoosmotic treatment. Colocalization between β1-integrin and both soluble FN and talin proved its active conformation (*Figure 7—figure supplement 2L–O*). We then analyzed the effect on adhesion and activation after silencing Tln1 and Tln2. Interestingly, Tln2-silencing did not significantly affect either hypoosmotic-induced β1-integrin activation or adhesion in Cav1KO MEFs (*Figure 7G–I and K–M* and *Figure 7—figure supplement 2K*). In contrast, simultaneous silencing of Tln1 and Tln2 in Cav1KO MEFs reduced hypoosmotic-induced β1 activation and reduced adhesion, rescuing the wild type phenotype (*Figure 7J and M* and *Figure 7—figure supplement 2K*, see also *Figure 7—figure supplement 2—source data 2 and 3*). The same result was observed in Cav1-silenced HeLa cells (*Figure 7—figure supplement 2—source data 2*). Surprisingly, Tln1/2 knockdown also affected integrin trafficking, as surface active β1-integrin was significantly reduced and consistently increased intracellularly in EEA-1 positive endosomes (*Figure 7—figure supplement 2P–W*). Given that the talin head domain binds and activates integrins (*Zhang et al., 2008*), we studied the ability of the wild type talin head to rescue adhesion in Tln1- and Tln2-silenced Cav1KO MEFs. As a control, we transfected cells with a DN talin-head mutant (L325R) that does not activate integrins or link them to the cytoskeleton (*Wegener et al., 2007*). Expression of the WT talin head, but not the mutant, rescued both the activation and the adhesion ability of Cav1KO MEFs (*Figure 7N–P and Q–R*). The talin head thus regulates adhesion and β1-integrin activation in Cav1KO MEFs.

## Discussion

Cellular mechanosensing is dependent on both integrins and caveolae, but how these two are coupled in this cellular response is poorly understood, especially during early steps of mechanoadaptation. Our MT-pulling experiment results show that Cav1KO MEFs adhere more strongly than wild type MEFs to FN-coated beads. This is consistent with the higher adhesion behavior of Cav1KO MEFs as compared to wild type MEFs in substrate adhesion assays. FN binds to integrins, and β1 is its main

receptor. The analysis of β1 distribution revealed an increased surface pool of active β1-integrin in Cav1KO MEFs. TIRF imaging and FRAP measurements indicated that β1-integrin has a more dynamic behavior in Cav1KO MEFs, appearing at and disappearing from the membrane plane faster than in wild type MEFs, through mechanisms distinct from diffusion. This prompted us to study endocytosis/ recycling rates. ELISA-based assays revealed no differences between Cav1KO and wild type MEFs in endocytosis rate at the time points analyzed, suggesting that Cav1 does not play a specific role. This finding contrasts with other reports showing that Cav1 is required for proper β1-integrin endocytosis (*Li et al., 2016*; *Du et al., 2011*); however, these observations were not made in a Cav1KO background and were performed at later time points than examined in our experiments (as we wanted to analyze specifically early integrin mechanosensing events, studied in the MT experiments). This time difference suggests that net integrin endocytosis at early time points could be compensated by other mechanisms in Cav1KO MEFs. Furthermore, and in accordance with previous reports , Cav1 could be restricting the endocytosis of a pool of β1-integrin in wild type MEFs (where it is mainly endocytosed through caveolae-dependent mechanisms) and becomes freed in Cav1KO MEFs to follow a different entry route. Indeed, our results indicate that β1-integrin is partially taken up in Cav1KO MEFs through CLIC/GEEC endocytosis, which provides a fast entry route, as reported for other cargoes (*Chaudhary et al., 2014*; *Cheng et al., 2010*; *Paul et al., 2015*). Interestingly, net endocytosis of active β1-integrins is higher than that of inactive pools (*Arjonen et al., 2012*; *Valdembri et al., 2009*). This is consistent with our data showing that Cav1KO MEFs display higher 9EG7 signal inside the cell and harmonizes with our findings that Cav1 genetic deficiency enhances CLIC endocytosis.

Differences between Cav1KO and wild type MEFs were apparent when we analyzed β1-integrin recycling rates at early time points after 10 min of endocytosis, being faster in Cav1KO MEFs. Interestingly, no differences in recycling were observed after 5 min of endocytosis, suggesting that in the presence of Cav1, β1-integrin becomes stabilized over time. This could potentially depend on cholesterol levels as loading wild type MEFs with cholesterol increases both surface and endosomal active β1-integrin availability, phenocopying Cav1KO MEFs phenotype. Raising endosomal cholesterol levels leads to increased exocytic activity in wild type MEFs, mirroring Cav1KO MEFs cholesterol accumulation, as work in our lab has previously demonstrated (*Albacete-Albacete et al., 2020*). This suggests that there might be a cholesterol threshold above which β1-integrin trafficking is dysfunctional, as it happens in the absence of Cav1. Furthermore, our results suggest that Cav1 is required for Rab11-dependent recycling of β1-integrin, which takes longer to reach the PM. Interestingly, in the absence of Cav1, β1-integrin accumulates preferentially at EEA-1-positive vesicles, following the Rab4-dependent 'short' recycling loop. This is consistent with our previous observations of Cav1 playing a role in determining the migration mode of fibroblasts (*Grande-García et al., 2007*), alternating between persistent or random migration. Faster recycling in Cav1KO MEFs can account for the elevated β1-integrin surface availability and therefore explain the resulting reinforcement. This interpretation is further supported by our previous findings that Cav1KO MEFs have a higher number of small focal adhesions that are rapidly turned over (*Grande-García et al., 2007*), since short but frequent integrin-ECM contacts would strengthen adhesion over time (ECM-coated beads in the MT experimental design). In the same report, we showed that Cav1KO MEFs have low Rho activity resulting in impaired actomyosin contraction (*Grande-García et al., 2007*), a condition that alters the overall cell mechanical response, as revealed in our traction force experiments. However, in the present study, Rho appears to be dispensable for observed differences in reinforcement. Consistently, previous reports have shown initial integrin adhesion in the absence of cytoskeleton connection, suggesting that early mechanosensing could be locally triggered (*Elosegui-Artola et al., 2014*; *Bakker et al., 2012*; *Changede et al., 2015*). This actomyosin-independent adhesion can derive from increased PM tension (*Wang et al., 2015*), which has also been shown to induce ligand-independent integrin activation (*Ferraris et al., 2014*). Cav1KO MEFs lack caveolae and are unable to buffer membrane tension upon mechanical stress (*Sinha et al., 2011*). In this condition, integrin recycling and activation at a lower force threshold might be facilitated by both faster exocytosis and easier switch of β1-integrin to its active conformation, as we observed upon hypoosmotic treatment. This increased sensitivity to membrane tension contributes, together with increased recycling, to the higher surface availability of active β1-integrin we observed in Cav1KO MEFs. Interestingly, we have also observed enhanced integrin recycling and activation in Cav1WT MEFs but after longer treatment times and with increased hypoosmotic pressure. These results suggest that caveolae membrane buffering is limiting

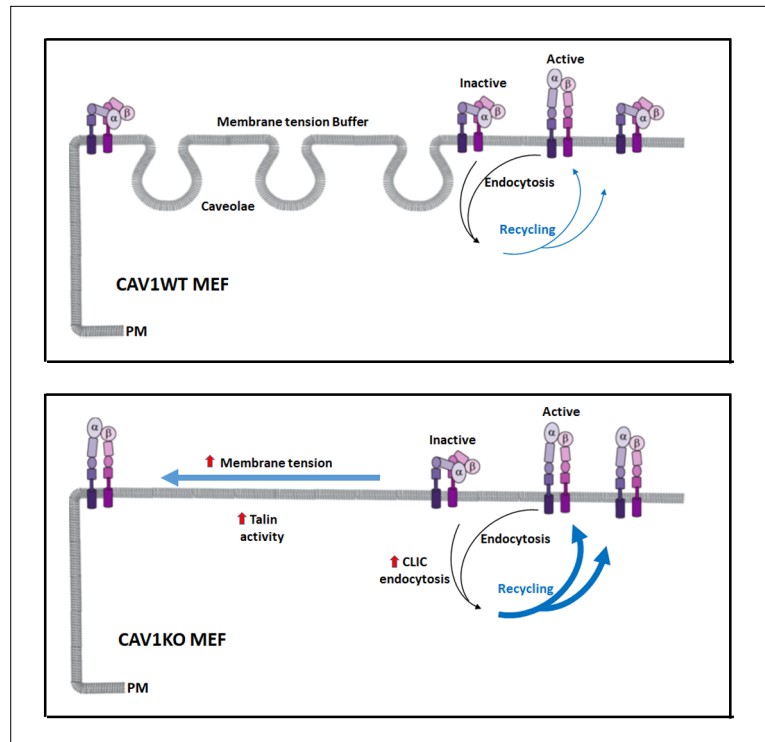

**Figure 8.** Caveolae adjust membrane tension to integrin mechanosensing by regulating integrin cycling and activation. Wild type mouse embryonic fibroblasts (MEFs) adapt to membrane tension changes through the buffer system of caveolae, driving a physiological integrin mechanosensing (in this case α5β1-integrin). In the absence of caveolae, dysregulation of this response leads to increased plasma membrane (PM) tension, which accelerates integrin recycling and switches integrin from the inactive forms to the active forms (close vs open conformation, respectively). Both increased β1-integrin recycling and activation is supported by increased talin activity in the absence of caveolae.

both integrin recycling and activation before a PM tension threshold is reached. Previous studies have shown that talin controls inside-out integrin activation (*Calderwood, 2004*; *Roca-Cusachs et al., 2013*) through its head domain (*Tadokoro et al., 2003*; *Zhang et al., 2008*). Interestingly, our results show that talin expression is required both for adhesion and for integrin activation in Cav1KO MEFs. It also seems to play a role in integrin recycling, as Tln1/2 knockdown altered integrin trafficking. Importantly, depletion of both talin paralogs does not affect initial cell spreading (*Zhang et al., 2008*). This is consistent with our observation that talin depletion does not affect initial attachment of Cav1KO MEFs to the substrate (*Figure 7M and 7P*, see Materials and methods for details), as cells were allowed in our adhesion assays to spread for 30 min prior to measurements. In contrast, cells were allowed to spread and form attachments for at least 48 hr after reverse siRNA transfection in experiments were hypoosmotic shock was induced, and—while still attached—Cav1KO/KD cells depleted for Talins were rounder and less spread (compare *Figure 7H* with *Figure 7J*) than control cells. Rescue experiments indicated that this situation is mainly supported by the talin head domain. This suggests the intriguing possibility that membrane tension could favor local integrin–talin-head interaction without force transmission in Cav1KO MEFs. These observations suggest that the cellular cytoskeleton might play only a minor role in the early cellular response to mechanical stress. A similar conclusion was recently prompted by the detection of mechanical-stress–induced integrin recruitment in the absence of significant cytoskeletal changes (*Elosegui-Artola et al., 2014*).

Our results indicate that caveolae impact both integrin surface availability, through adjusting recycling, and integrin activation, through membrane tension regulation and talin activity. Cav1KO MEFs, lacking this control mechanism, show both increased β1-integrin recycling and surface activation. This results in dysregulated early mechanosensing (*Figure 8*) and subsequent inability to properly sense environmental stiffness, a situation known to impact tumorigenesis (*Lin et al., 2015*) and stem

cell differentiation (*Li et al., 2016*). Mechanobiology is an emerging field, essential to understand how cells and tissues adapt to their environment in health and disease (*Zaidel-Bar, 2017*). Our study provides novel insight on the role of caveolae in early integrin mechanosensing, revealing a new layer of complexity at the interface of physics and biology.

## Materials and methods

### Cloning, cells, and reagents

Caveolin-1 flag was excised from pCDNA3.1 Cav1 with BamH1/EcoR1, klenow treated, and ligated into the klenow blunt-ended EcoR1 site of the GFP-expressing retroviral vector MIGR1. PTRF was excised from pIRES2-cavin1 EGFP with BglII/BamH1 and ligated into the BglII site of MIGR1. The C-terminally EGFP-tagged β1-integrin was developed by Prof. Martin Humphries (University of Manchester, UK) and described elsewhere (*Parsons et al., 2008*) and was requested through the Addgene public repository under number #69767.

All cells were cultured at 37°C and 5% $CO_2$ in DMEM (Thermo Fisher Scientific) supplemented with 10% fetal bovine serum (FBS) and 1% penicillin and streptomycin. Cav1KO MEFs were kindly provided by Michael Lisanti (Institute of Cancer Sciences, Manchester). All cell cultures were routinely checked for mycoplasma contamination. To deplete talin 2, cells were transfected with a plasmid encoding talin 2 shRNA and puromycin resistance (*Zhang et al., 2008*). Puromycin (2 µg/ml) was added to the cells 24 hr after transfection and maintained for 4 days to select transfected cells. To deplete talin 1, talin 2 shRNA-stable cells were transfected with Tln1 esiRNA (EMU083531, Sigma Aldrich) or, for rescue experiments with the talin head, with Tln1 siRNA (Silencer Select, Life technologies), which does not target the talin head domain. For rescue experiments, talin 2 shRNA-stable cells were co-transfected with Tln1 siRNA and EGFP-talin 1 head (Addgene plasmid no. 32856) or EGFP-talin 1 L325R (the mutant version, kindly provided by M. Ginsberg, UC San Diego, USA).

The following primary antibodies were used: mouse monoclonal anti-transferrin receptor (H68.4, Catalog # 13–6800), rat monoclonal anti-mouse total β1-integrin (clone MB1.2, MAB1997 Millipore); rat monoclonal anti-mouse β1-integrin, activated (clone 9EG7, BD Pharmingen); Alexa 488 conjugated anti-integrin β1, activated (clone HUTS-4, MAB2079-AF488 Millipore); rabbit polyclonal anti-mouse PTRF (Abcam); rabbit monoclonal anti-mouse caveolin-1 (Cell Signaling, #3238); mouse monoclonal anti-tubulin (Abcam, clone DM1A); rabbit monoclonal anti-CD147 (Invitrogen, clone JF1-045), mouse monoclonal anti-EEA-1 (BD transduction, clone 14); mouse monoclonal anti-LBPA (Echelon Z-SLBPA); mouse monoclonal anti-p190RhoGAP (Upstate, clone D2D6, 1:1000); rabbit monoclonal anti-mouse Caveolin-1 (Cell signaling, 1:1000); mouse monoclonal anti-alpha tubulin (ab7291, Abcam, 1:10.000); mouse monoclonal anti-Talin (Sigma Aldrich, clone 8d4, 1:200); and mouse anti-CD44 (clone 5035–41.1D, Novus Biologicals). FN and LDLs were obtained by purification from blood donors and were conjugated with FITC (Thermo Fisher) or used directly, respectively. U18666A was from sigma (U3633). Secondary antibodies were Alexa Fluor–488 goat anti-rat (Thermo Fisher Scientific); Alexa Fluor–647 goat anti-rat (Thermo Fisher Scientific); Alexa Fluor–488 phalloidin (Thermo Fisher Scientific); HRP-linked anti-biotin from Cell Signaling (#7075); and Alexa Fluor–647 phalloidin (Thermo Fisher Scientific). EZ-Link SulfoNHS-SS-biotin was from Thermo Fisher Scientific (D21331), 2-mercaptoethanesulfonic acid (MESNA) and iodoacetamide from Sigma Aldrich (63707 and I1149), the Cdc42 inhibitor ML141 from Tocris Bioscience, and Alexa Fluor 647-Dextran from Thermo Fisher Scientific (D22914). Silencing of p190RhoGAP was as previously described (*Grande-García et al., 2007*).

### PM fractionation and western blot analysis

MEFs were processed for PM isolation as described (*Smart et al., 1995*). All steps were carried out at 4°C. Cells were first washed with cold-PBS 1× and pelleted by centrifugation at 14000×g for 5 min. Cells were then manually homogenized with 20 strokes of a PTFE head Tissue homogenizer (VWR) and centrifuged at 1000×g for 10 min. The post-nuclear supernatant was collected and layered atop a 30% Percoll column. After centrifugation of the Percoll column at 84,000×g for 30 min, the PM fraction was a visible band around 5.7 cm from the bottom of the centrifuge tube, was collected, further centrifuged at 105,000 g for 1 hr to remove Percoll, separated by SDS-PAGE, and finally analyzed by western blot. Samples were immunoblotted with rabbit monoclonal anti-mouse caveolin-1 and anti-mouse

monoclonal anti-human transferrin receptor (clone H68.4, Invitrogen 136800, RRID:AB_2533029) as a loading control and a marker for PM fraction.

Total cell lysates were separated by SDS-PAGE and analyzed by western blot with rabbit monoclonal anti-mouse Caveolin-1 and rabbit polyclonal anti-mouse PTRF (Abcam), with mouse monoclonal anti tubulin used as the loading control. Secondary antibodies were goat anti-mouse 800 and goat anti-rabbit 680. All membranes were scanned with the Odyssey imaging system (Li-COR).

## Electron microscopy

MEFs were processed for EM using standard procedures. Briefly, cells were fixed for 1 hr with 2.5% glutaraldehyde in 100 mM cacodylate buffer, pH 7.4, and then post-fixed for 3 hr with 1% osmium tetroxide in 100 mM cacodylate buffer, pH 7.4. The samples were dehydrated with acetone, embedded in Epon and sectioned. Ruthenium red (1 mg/ml) was added during fixing and post-fixing to stain the PM.

## Confocal and ground state depletion microscopy

Confocal images were obtained with an LSM 700 inverted confocal microscope (Carl Zeiss) fitted with a 63×1.4 NA objective and driven by Zen software (Carl Zeiss). Superresolution imaging was performed with a GSD-TIRF microscope (Leica Microsystems). Samples were prepared according to standard procedures indicated by Leica Microsystems. The primary antibody was rabbit monoclonal anti-mouse caveolin-1 (1:100), and Alexa Fluor 647 Fab1 fragment goat anti-rabbit (Jackson Immunoresearch; 1:100) was used as the secondary to further improve spatial resolution.

## MTs and reinforcement measurements

### Bead coating

Carboxylated magnetic beads (Invitrogen) were mixed in a solution containing 500 µl 0.01 M sodium acetate pH 5, 0.75 mg Avidin (Invitrogen), and 4 mg EDAC (Sigma). Beads were incubated for 2 hr at room temperature (RT) and then washed in PBS and further incubated for 30 min in 1 ml 50 mM ethanolamine (Polysciences). The beads were then washed three times in PBS and left in PBS on a cold room rotator.

### Force measurements

MTs experiments were performed as described (*Roca-Cusachs et al., 2009*). Briefly, carboxylated 3 µm magnetic beads (Invitrogen) were coated with biotinylated pentameric FN7-10 or ConA (Sigma Aldrich) mixed 1:1 with biotinylated BSA. For measurements, cells were first plated on coverslips coated with 10 µg/ml FN (Sigma) in Ringer's solution (150 mM NaCl, 5 mM KCl, 1 mM CaCl$_2$, 1 mM MgCl$_2$, 20 mM HEPES, and 2 g/L glucose, pH 7.4) for 30 min. FN-coated beads were then deposited on the coverslips and allowed to attach to the cells. The tip of the MTs device was then used to apply a force of 1 nN for 2 or 3 min to beads attached to cell lamellipodia. The apparatus used to apply force to the magnetic beads was as previously described (*Tanase et al., 2007*). The system was then mounted on a motorized 37°C stage on a Nikon fluorescence microscope. DIC images and videos were recorded with a 60× objective linked to a CCD camera at a frequency of 250 frames/s.

## Magnetic twisting

Cells seeded on FN-coated coverslips were subjected to magnetic twisting as previously described (*Trepat et al., 2004*). After twisting, cells were fixed for 20 min in 4% PFA and then stained with 9EG7 antibody without permeabilization.

## Silica bead coating and staining

Carboxylated silica beads (3 µm, Kisker Biotech) were prepared as described above except that ConA-coated beads were incubated with biotinylated BSA previously labeled with an Alexa Fluor 555 protein labeling kit (Invitrogen). Unlabeled beads (FN-coated) and labeled beads (ConA-coated) were mixed in the same proportion (1:1). Cells were allowed to spread for 15 min and then fixed with 4% PFA, permeabilized with 0.1% Triton-X 100, and incubated at room temperature for 1 hr with the indicated antibodies. Only beads at the cell periphery were analyzed (excluding cells in the ectoplasm-endoplasm border zone). To quantify fluorescence intensity, a 10-pixel-diameter ring was

drawn around each selected bead using ImageJ. Mean fluorescence per area was normalized and plotted.

## Traction force microscopy

Traction force was measured as previously described (*Elosegui-Artola et al., 2014*). Briefly, cells were seeded on polyacrylamide gels incorporating embedded fluorescent nanobeads. Cells were imaged by phase contrast and embedded nanobeads by fluorescence. Cells were then trypsinized, and bead position images were acquired in the relaxed gel state. Comparison of bead positions in gels with and without cells was used to obtain a gel deformation map (*Serra-Picamal et al., 2012*; *Bazellières et al., 2015*). Images were obtained with a Nikon Eclipse Ti inverted microscope fitted with a 40× objective (numerical aperture = 0.6).

## Adhesion assay

Cell adhesiveness was assessed by seeding MEFs on 96-well plates coated with FN or ConA (both at 5 µg/ml) and incubating at 37°C for 30 min. Wells with no coating were included as negative controls. Cells were then fixed with methanol and stained with crystal violet (Sigma Aldrich). Wells were washed thoroughly to remove excess dye and were finally eluted with a mixture of 50% ethanol and 50% 0.1 M sodium citrate (pH 4.2). The absorbance was read at 595 nm.

## Fluorescence recovery after photobleaching

Cav1KO and wild type MEFs were transfected with the β1-GFP expression vector. Two pre-bleached events were acquired before bleaching by stimulation with the Nikon scanner at 488 nm. Fluorescence recovery was monitored continuously until the intensity plateaued (approximately 1.5 min). Fluorescence during recovery was normalized to the pre-bleach intensity. Cells were cultured in DMEM (Thermo Fisher Scientific) supplemented with 10% FBS, and 1% penicillin, and streptomycin.

## Endocytosis/recycling assay

β1-integrin kinetics were analyzed after biotin labeling of cell-surface integrins followed by a capture ELISA-based assay, using a modification of a previously described protocol (*Li et al., 2016*).

### Cell-surface integrin biotinylation

Wild type or Cav1KO MEFs ($5\times10^5$) were seeded on five (endocytosis) or four (recycling) matrigel-coated plates* (labeled total, 0 min, 2 min, 5 min, and 10 min for endocytosis and 0 min, 1 min, 3 min, and 5 min for recycling). The cells were incubated in complete DMEM (Thermo Fisher Scientific) for 2 hr at 37°C, which was the shortest spreading time for cells to stand the assay conditions and also matches the MTs measurement experimental time frame. The plates were then placed on ice, washed twice with ice-cold PBS and incubated for 40 min at 4°C with 0.25 mg/ml of EZ-Link SulfoNHS-SS-Biotin in Hank's balanced salt solution (Sigma Aldrich). After two further washes with ice-cold PBS, the plates were labeled for the appropriate time points and processed as described below. *Matrigel was used instead of FN to mimic a more physiological environment. While different coatings can clearly affect integrin trafficking, matrigel contains, among other extracellular components, collagen and certain levels of FN, which we have shown to increase Cav1KO adhesion as compared to wild type MEFs (see *Figure 3—figure supplement 1B and C*). Importantly, cells were allowed to spread for 2 hr before starting the assay in complete medium, thus ensuring an additional supply of FN.

### Endocytosis

The 2, 5, and 10 min plates were incubated at 37°C with 2 ml of pre-warmed DMEM (without FBS) for the indicated times. The total and 0 min plates were placed on ice with 2 ml ice-cold DMEM (without FBS). All plates except total were then washed twice with ice-cold PBS and incubated for 40 min at 4°C with MESNA-containing buffer (Sigma Aldrich) to remove remaining surface-associated biotin. All the plates were then washed twice with ice-cold PBS and incubated with iodoacetamide for 10 min at 4°C. After washing again with ice-cold PBS, cells were lysed and processed for ELISA.

## Recycling

The 0, 1, 3, and 5 min plates were incubated at 37°C with 2 ml of pre-warmed DMEM (without FBS) for 10 min (to allow time for endocytosis). The plates were then washed twice with ice-cold PBS and incubated for 40 min at 4°C with MESNA-containing buffer to remove remaining surface-associated biotin. The 1, 3, and 5 min plates were incubated at 37°C with 2 ml of pre-warmed DMEM (without FBS) for the indicated times; the 0 min plate was placed on ice with 2 ml ice-cold DMEM (without FBS). At the end of the incubation, all plates were washed twice with ice-cold PBS and incubated again with MESNA-containing buffer to remove biotin-labeled integrins that had recycled to the PM. Plates were finally washed with ice-cold PBS, incubated with iodoacetamide for 10 min at 4°C, lysed, and processed for ELISA.

## ELISA-based assay

96-well ELISA plates were coated overnight at 4°C with anti-mouse total β1-integrin (Millipore) or anti-mouse β1-integrin, activated (BD Pharmingen). The plates were then washed three times with solution A (0.02% Tween-20 in PBS), blocked for 1 hr at room temperature with solution B (0.02% Tween-20, 1% BSA in PBS), and incubated with cell lysates from endocytosis or recycling assays for 2 hr at room temperature or overnight at 4°C. After washing three times with solution A, the plates were incubated for 1 hr at room temperature with anti-biotin HRP-linked antibody. Plates were then washed three more times with solution A, and integrin was detected by TMB reaction (Sigma Aldrich). Endocytosis results were normalized by dividing with the signal from Total wells; graphs represent the progressive increase in the amount of total or active β1-integrin. Recycling results were normalized by dividing with the signal from 0 min wells; graphs represent the β1-integrin remaining inside cells, so that negative curves indicate an increase in the recycling rate.

To determine total cell-surface integrin, cells were processed as in the total plates. For Rab DNs and Tln siRNA experiments, cells were plated 48 hr before the experiment avoiding re-plating to prevent losing lesser adherent cells. For cholesterol loading experiments, cells were treated with U18666A 2 µg/ml or LDL 100 µg/ml for 24 hr before surface integrin quantification.

## Total internal reflection fluorescent microscopy videos

TIRF microscopy was performed with a Leica AM TIRF MC microscope. TIRFm movies were acquired with a 100 X_1.46 NA oil-immersion objective at 488 nm excitation and an evanescent field with a nominal penetration depth of 100 nm. Images were collected with an ANDOR iXon CCD at 300 ms per frame. Quantification of TIRF videos show normalized fluorescence integrated density (IntDen) over frames (*Figure 4—figure supplement 1C–1E*). Graph represents the mean of the difference between normalized fluorescence IntDen of adjacent frames (frame$_x$–frame$_{x-1}$) (*Figure 4—figure supplement 1F*). IntDen was calculated as in the following formula: IntDen = Raw IntDen (sum of pixel values in selection) × (area in scaled units)/(area in pixels), which was then normalized by the IntDen mean of all frames analyzed.

## Hypoosmotic treatment

Cells were cultured for the indicated time points at 37°C in DMEM diluted as indicated with distilled water. The cells were then immediately fixed for 15 min by adding an equal volume of 8% PFA (yielding a final PFA concentration of 4%) and then stained with 9EG7 antibody without permeabilization.

## Endocytosis experiments

For dextran endocytosis: cells were first incubated for 1 hr at 4°C (to prevent endocytosis) with β1-Alexa 488 conjugated antibody, activated. Cells were then washed twice with PBS and incubated for 3 min at 37°C with 1 mg/ml Alexa Fluor 647-Dextran (Invitrogen, REF D22914) without pre-incubation or acid stripping. Cells were then fixed with 4% PFA and analyzed by confocal microscopy. Colocalization was analyzed using the plugin Coloc 2 (Fiji *Schindelin et al., 2012*).

For caveolar uptake, cells were treated, when indicated, with genistein 200 uM for 2 hr before incubation with BODIPY-LacCer 5 uM for 1 hr at 4°C followed by 3 min endocytosis at 37°C. BODIPY-LacCer (Invitrogen, REF B34402) remaining at the PM was then removed by back exchange at 4°C following a previous protocol (*Martin and Pagano, 1994*). Cells were then fixed with 4% PFA, stained, and analyzed by confocal microscopy.

For CLIC endocytosis, cells were treated, when indicated, with ML141 inhibitor 10 uM (Tocris, REF 4266) for 30 min before incubation with anti-active 1-Alexa 488 antibody and anti-CD44 antibody for 1 hr at 4°C followed by 3 min endocytosis at 37°C. Cells were then acid stripped to remove surface staining, fixed with 4% PFA, stained, and analyzed by confocal microscopy.

For clathrin endocytosis cells were incubated with anti-active 1-Alexa 488 antibody and Tnf-568 (Invitrogen, REF T23365) for 1 hr at 4°C followed by 3 min endocytosis at 37°C. Cells were then acid stripped to remove surface staining, fixed with 4% PFA, stained, and analyzed by confocal microscopy. Colocalization was analyzed using the plugin Coloc 2 (Fiji ).

## Extracellular staining

To analyze cell-surface β1-integrin, cells were fixed for 20 min with 4% PFA and then stained with 9EG7 antibody without permeabilization. Alternatively, to rule out any possible PM disruption due to PFA fixation, cells were also incubated with 9EG7 antibody at 4°C for 1 hr as indicated in *Figure 7—figure supplement 2A–J*.

## Statistical analysis

Data are presented as mean ± SEM unless otherwise indicated. Mean values were compared by two-tailed paired Student t-test unless otherwise indicated. Differences were considered statistically significant at $p < 0.05$ (*), $< 0.01$ (**), and $< 0.001$ (***).

## Acknowledgements

We thank Dr. Miguel Sánchez for critical reading of the manuscript and Simon Bartlett for scientific editing. We also thank Verónica Labrador Cantarero and Antonio M Santos Beneit from Microscopy Unit (CNIC) for macro development and video editing and Dr. Martin Humphries (The University of Manchester), Dr. Michael Lisanti (Institute of Cancer Sciences, Manchester), Dr. M Ginsberg (UC San Diego, USA), and Dr. Cristina Clemente Toribio for kindly providing reagents and cells.

This project received funding from the European Union Horizon 2020 Research and Innovation Programme through Marie Sklodowska-Curie grant 641639; grants from the Spanish Ministry of Science and Innovation (MCIN/AEI/10.13039/501100011033): SAF2014-51876-R, SAF2017-83130-R cofunded by "ERDF A way of making Europe", PID2020-118658RB-I00, PDC2021-121572-100 cofunded by "European Union NextGenerationEU/PRTR", CSD2009-0016, and BFU2016-81912-REDC; and the AECC (Asociación Española Contra el Cáncer) foundation (PROYE20089DELP) all to MAdP. MAdP is member of the Tec4Bio consortium (ref. S2018/NMT¬4443; CAM/FEDER, Spain), co-recipient with PR-C of grants from Fundació La Marató de TV3 (674 /C/2013 and 201936-30-31), and coordinator of a Health Research consortium grant from Fundación Obra Social La Caixa (AtheroConvergence, HR20-00075). The CNIC Unit of Microscopy and Dynamic Imaging is supported by FEDER "Una manera de hacer Europa" (ReDIB ICTS infrastructure TRIMA@CNIC, MCIN). The CNIC is supported by the Instituto de Salud Carlos III (ISCIII), the MCIN and the Pro CNIC Foundation, and is a Severo Ochoa Center of Excellence (grant CEX2020-001041-S funded by MICIN/AEI/10.13039/501100011033).

## Additional information

### Competing interests

Dácil María Pavón: is affiliated with Allergy Therapeutics S.L. The author has no financial interests to declare. The other authors declare that no competing interests exist.

### Funding

| Funder | Grant reference number | Author |
|---|---|---|
| European Union Horizon 2020 Research and Innovation Programme | Marie Sklodowska-Curie grant 641639 | Miguel A del Pozo |

| Funder | Grant reference number | Author |
|---|---|---|
| Asociación Española Contra el Cáncer Foundation | PROYE20089DELP | Miguel A del Pozo |
| Spanish Ministry of Economy, Industry and Competitivenes | SAF2014-51876-R | Miguel A del Pozo |
| Spanish Ministry of Economy, Industry and Competitiveness | SAF2017-83130-R | Miguel A del Pozo |
| Spanish Ministry of Economy, Industry and Competitiveness | CSD2009-0016 | Miguel A del Pozo |
| Spanish Ministry of Science and Innovation | PID2020-118658RB-I00 | Miguel A del Pozo |
| Spanish Ministry of Science and Innovation | PDC2021-121572-100 | Miguel A del Pozo |
| Comunidad Autónoma de Madrid | S2018/NMT¬4443 | Miguel A del Pozo |
| Fundació la Marató de TV3 | 201936-30-31 | Pere Roca-Cusachs |
| Ministerio de Ciencia e Innovación | CEX2020-001041-S | Miguel A del Pozo |

The funders had no role in study design, data collection and interpretation, or the decision to submit the work for publication.

## Author contributions

Fidel-Nicolás Lolo, Conceptualization, Data curation, Formal analysis, Supervision, Validation, Investigation, Methodology, Writing – original draft, Writing – review and editing; Dácil María Pavón, Araceli Grande-García, Sara Sánchez, Data curation, Formal analysis, Investigation; Alberto Elosegui-Artola, Valeria Inés Segatori, Formal analysis, Investigation; Xavier Trepat, Supervision, Methodology; Pere Roca-Cusachs, Conceptualization, Formal analysis, Methodology; Miguel A del Pozo, Conceptualization, Resources, Supervision, Funding acquisition, Writing – original draft, Project administration, Writing – review and editing

## Author ORCIDs

Fidel-Nicolás Lolo http://orcid.org/0000-0003-1635-4770
Araceli Grande-García https://orcid.org/0000-0003-2619-5013
Xavier Trepat http://orcid.org/0000-0002-7621-5214
Pere Roca-Cusachs http://orcid.org/0000-0001-6947-961X
Miguel A del Pozo http://orcid.org/0000-0001-9077-391X

## Decision letter and Author response

Decision letter https://doi.org/10.7554/eLife.82348.sa1
Author response https://doi.org/10.7554/eLife.82348.sa2

## Additional files

### Supplementary files
• MDAR checklist

### Data availability
Raw data of all figures is included as excel files.

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
