## [Editor Report]

This valuable cell biological study uses magnetic tweezers to explore how integrins and caveolae interact to regulate mechanosensing. The authors describe a convincing link between the presence of caveolae and the trafficking of integrins between the cell surface and intracellular compartments to control plasma membrane tension.

---

## [Decision Letter]

**Decision letter after peer review:**

[Editors’ note: the authors submitted for reconsideration following the decision after peer review. What follows is the decision letter after the first round of review.]

Thank you for submitting your work entitled "Cav1/caveolae couple mechanical stress to integrin recycling and activation" for consideration by *eLife*. Your article has been reviewed by 3 peer reviewers, and the evaluation has been overseen by a Reviewing Editor and a Senior Editor. The reviewers have opted to remain anonymous.

Our decision has been reached after consultation between the reviewers. Based on these discussions and the individual reviews below, we regret to inform you that your manuscript will not be considered further for publication in *eLife*.

There is no question that the role of caveolae in the mechanobiology of the plasma membrane and crosstalk between caveolae and integrins is of broad interest and importance. However, as you will see from the reviewer comments, there was concern that potential trafficking defects had not been connected to integrin activation and that the Rab4 trafficking experiments would require further work. Overall the reviewers felt that the story lacked a mechanistic link and many of the conclusions were not adequately documented to support the conclusions of the study.

We realize that this decision will be disappointing but we hope that the referee comments will be helpful to you as you plan your next steps. If you are able to address all of the concerns with significant additional experiments, a new manuscript could be evaluated in the future.*Reviewer #1:*

Caveolae are now widely recognized for their roles in the mechanobiology of the plasma membrane of cells submitted to mechanical stress. In the current manuscript, the authors have studied the crosstalk between caveolae and another mechanosensitive group of proteins, the integrins. Using as a model system mouse embryonic fibroblasts (MEFs) that are deleted for the caveolin-1 gene, it is described that cell adhesion to fibronectin-coated surfaces is increased in the absence of caveolae. The underlying reason for this is identified in the increased presence of the active conformation of beta1 integrin at the cell surface, which results from 2 phenomena: increased endocytic recycling, and increased activation upon mechanical stress.

The overall area of research on mechanobiology in relation with membrane trafficking that is addressed in the current study is of high interest to a general readership in the life sciences. The study is based on a rich arsenal of relevant techniques, and represents a substantial amount of work. Some techniques such as cell adhesion stiffness measurements based on magnetic bead oscillation or traction force microscopy are really quite elegant. The manuscript is well written, and pleasant to read.

Possibly the most important limitation of the current study is that it remains descriptive on the molecular mechanism aspects. It is clear that the absence of Cav1/caveolae has a number of effects that are clearly defined in the current study. The molecular wiring that underlies these effects remains unexplored at this stage. A few examples: In the absence of Cav1/caveolae, endocytic recycling of beta1 integrin passes from Rab11-dependent slow recycling to Rab4-dependent fast recycling – by which molecular mechanism does Cav1/caveolae connect to the endosomal machinery? Mechanical stress increases beta1 activation in Cav1KO MEFs – by which molecular mechanism does Cav1/caveolae limit this activation in wild-type conditions? Talin is important in the context of this mechanical stress-controlled activation reaction – by which molecular mechanism does Cav1/caveolae interact with talin? Having said this, I still believe that the manuscript will contribute in a substantial manner to the dynamic field of membrane biology research. The following points should be clarified, though.

Lines 262-269: The faster beta1 integrin recycling phenotype in Cav1KO MEFs is seen only after 10 min of endocytosis, and not when recycling is measured after 5 min of endocytosis. The authors suggest that "beta1-integrin is stabilized in the presence of Cav1 after 10 minutes endocytosis". How would such stabilization work? The authors most likely don't have the experimental response to this question at this stage. However, it would be helpful if some ideas could be discussed on the molecular mechanisms by which this would work.

Lines 315-317: In the paragraphs that precede these lines, the authors present experiments based on the small molecule Cdc42 inhibitor ML141 that lead to the conclusion that in Cav1KO MEFs, part of endocytic uptake occurs by the CLIC/GEEC pathway, while this does not appear to be the case in MEFs that express Cav1. It is concluded that in "wild-type" conditions, part of uptake occurs through caveolae, and that this Cav1-dependent internalization would be compensated by the CLIC/GEEC pathway in the absence of Cav1/caveolae. Can other interpretations be excluded? For example, wouldn't one observe the same phenotype if Cav1/caveolae were to inhibit uptake of a fraction of the beta1 integrin molecules that once liberated in the Cav1KO condition would now be internalized by the CLIC/GEEC pathway? The most direct way to address this point would be electron microscopy to provide ultrastuctural images of the uptake structures in which beta1 integrin is found in the different experimental conditions. To the least, alternative interpretations of the data should be discussed.*Reviewer #2:*

This manuscript describes the role of Caveolin-1 (Cav1) in integrin recycling using Cav1 knockout MEFs and links integrin endocytosis and recycling to cell mechanosensing and adhesion. I find the subject interesting and, in light of the role of Cav1 and caveolae in the cellular response to mechanical stress, timely and relevant. That being said I find the manuscript to be a compilation of interesting results that lack mechanistic connection between them. We are provided with interesting data on: (1) the role of Cav1 in integrin mechanosensing using an elegant magnetic tweezer approach; (2) Cav1 regulation of integrin endocytosis and recycling but with no direct link to mechanosensing or stretch; (3) the role of talin in regulating CAV1-independent integrin activation. It seems as though Cav1 is altering integrin dynamics and response to mechanical stretch which is quite interesting. While the schematic in figure 8 highlights the Cav1-dependent changes reported in the paper, mechanistic connections between them need to be clarified and I have many questions which I outline below.

1. "Cav1/caveolae" Cav1 and caveolae are not the same thing and it cannot be assumed that effects observed in Cav1 KO cells are necessarily attributed to loss of caveolae. I am concerned about use of this term throughout the paper (even the title) and suggest that it would be important to define the specific role of caveolae in the processes described using cell lines expressing Cav1 but not PTRF. Indeed, the only data linking caveolae flattening to the effects shown is Figure 7 and is interesting in that it suggests that lack of membrane buffering by caveolae induces an integrin response. However, to definitively show that this is caveolae dependent and not Cav1 dependent it would be important to use cells expressing Cav1 but not caveolae.

2. The studies on b-integrin endocytosis switch to a CLIC pathway in Cav1 KO is interesting and supports a role for Cav1 as an inhibitor of CLIC endocytosis. It would be important to provide more evidence for CLIC endocytosis than inhibition of Cdc42. Is b-integrin cointernalized with CD44? Is this pathway CD44- or raft-dependent? Can the effects of Cav1 on integrin mechanosensing be attributed to CLIC endocytosis of integrin? How much of total surface integrin is internalized via this pathway and not clathrin or caveolin pathways? Is CLIC (and fluid phase) endocytosis generally upregulated in CAV1 KO cells? If the endocytosis effects are most clearly seen at early times of cell spreading how does this relate to the mechanosensing experiments done on spread cells?

3. I am also a little confused as to how the authors envisage Cav1 regulating integrin recycling. Is Cav co-internalized with b-integrin and stabilizes it in endosomes, slowing recycling? Is this occurring in caveosomes? If this is the case it would be important to show it. Can the TIRF videos be quantified to support the ELISA data?

4. If the issues related to Cav1 vs caveolae mentioned in 1 are addressed, caveolae membrane buffering may be limiting integrin activation and an integrin-dependent tensive response. This is very interesting. Caveolae membrane buffering is thought to be an early response to mechanical stress so shouldn't WT cells show the same effect at longer times or with increased hypoosomotic pressure? Can this be tested? And what about other mechanical stressors?

5. Finally, we are shown in Figure 7 that Talin regulates adhesion and b-integrin activation in KO cells. Is Talin required for the increased integrin dynamics shown in previous figures?*Reviewer #3:*

This paper from Lolo et al. described experimentation aimed at determining how caveolin and caveolae influence integrin function. The use mouse embryonic fibroblasts (MEFs), and caveolin null MEFs that have, or have not been rescued with exogenously expressed caveolin. They first use magnetic tweezers to show that knockout of caveolin influences integrin mechanical behaviour. They then used a surface biotinylation assay to look at the endocytosis and endocytic recycling of beta1 integrin. They found that caveolin null MEFs had unaffected levels of integrin endocytosis, but that they recycled bet1 integrin more rapidly than wild-type MEFs. They also show that a small pool of beta1 integrin is internalised through a cdc42-dependent pathway and attribute this to the clathrin independent carrier (CLIC) pathway, but it is unclear from these data whether this is influenced by caveolin. The authors proceed from this to use immunofluorescence colocalisation studies and dominant negative mutants of Rab GTPases to determine that caveolin null MEFs may have altered integrin trafficking, possibly via a Rab4-dependent route. Finally, the authors show that hypoosmotic shock (to put the plasma membrane under tension) leads to increased integrin activation in caveolin null MEFs and that this may be talin-dependent.

In general this paper describes a series of observations which are not sufficiently well-connected to support the synthesis that the authors assemble for how caveolin may regulate integrin trafficking and function. Moreover, much of the experimentation and the way that the results are presented are neither convincing nor of the standard necessary for publication in *eLife*.

1. Figure 4. The authors have used a surface biotinylation method to show that integrin recycling is different in wild-type and caveolin null MEFs. These assays need to be performed with the rescue cells too.

2. The authors pursue their recycling results using immunofluorescence (Figure 6) to try to infer that differences in recycling, shown in Figure 4 are Rab4-dependent. I do not find any of the data in Figure 6 convincing. It is not clear to me why the various colocalisations tested represent recycling of integrins. Moreover, how can one look at colocalisation with dominant negative mutants of Rabs (which are known to localise aberrantly) and conclude that a particular cargo is in any given compartment?

3. How are the data in Figure 5 consistent with those in Figure 4? The authors show in Figure 4 that integrin endocytosis is not affected by caveolin knockout. In Figure 5, they show that an inhibitor of the CLIC pathway (although they do not validate its efficacy or specificity) slightly inhibits integrin endocytosis. They also show that fluid-phase endocytosis is increased in in caveolin null MEFs and infer that integrins are internalised through this pathway. But, in Figure 4 integrin endocytosis is not different between caveolin wild-type and null fibroblasts. Thus, I am completely at a loss to interpret the authors' concluding statement which reads: 'Thus, in the absence of Cav1/caveolae, early endocytosis of b1-integrin occurs at least in part through fluid-phase uptake, which provides an alternative entry route that would compensate for lack of Cav1-dependent internalization.'

4. There are no attempts, as far as I can see, to link the alterations in integrin trafficking to their activation. I would have thought that, having shown that integrin trafficking, their mechanical properties and their activation-state (9EG7 binding) are different between caveolin null and wild-type cells, the authors would want to attempt to determine which of these events are mechanistically connected. At least, for *eLife* I would expect some of these questions to be addressed.

5. The 9EG7 staining data presented in Figure 7 are really not convinding. I do not see how the authors can quantify images such as those shown in Figure 7D. The staining just looks like a splodge of fluorescence implying some kind of aggregation, or gross disruption of the cell membrane. Also, some of this staining looks to be at the cell surface (Figure 7L), and some is clearly intracellular (Figure 7I), so how can this necessarily reflect active, talin-bound integrin that is competent to bind ligand?

6. The presentation of much of the data in this manuscript is not of the standard that is necessary for acceptance in *eLife*. Many of the axes are not labelled (such as the y-axes in Figure 4F – I), or there are labels/designations that are unclear; for instance, what does 'recycling by timepoint 0 after 10 min endocytosis' mean (Figure 4G)? Also, there are metrics which are not described at all. For instance, I cannot seem to find in the manuscript (or in reference 19) what 'reinforcement increment' is and how it is obtained. Is it the same as 'relative stiffening' in reference 19?

7. The hugely increased 'reinforcement increment' of ConA beads in Figure 2H over fibronectin beads in Figure 2G doesn't seem to square with the authors' assertion that integrins couple the beads to the cytoskeleton to restrict their movement. It looks to me that these data indicate that the ConA beads are much more tightly associated with the cytoskeletal machinery than the fibronectin-coated ones.

8. To support their conclusion that Rho is not involved in Cav1 knockout mediated reinforcement (Figure 3C) don't the authors need to show that the reinforcement increment still increases when one compares Cav1+/+;p190KD and Cav1-/-; p190KD cells? Curiously, the authors have not shown this.

[Editors’ note: further revisions were suggested prior to acceptance, as described below.]

Thank you for submitting the paper "Caveolae couple mechanical stress to integrin recycling and activation" for consideration by *eLife*. Your article has been reviewed by 3 peer reviewers, and the evaluation has been overseen by a Reviewing Editor and a Senior Editor.

Comments to the Authors:

We realize that you went to great effort to try to improve a prior manuscript but sadly must write to say that, after consultation with the reviewers, we have decided that this manuscript will not be considered further for publication by Life.

*Reviewer #1 (Recommendations for the authors):*

This paper proposes to link caveolin-1 regulation of the response to mechanical stress to beta1-integrin recycling. While some of the data are interesting, in particular, the Cav1-dependent response of cells to magnetic beads, other aspects of the paper are less than convincing. B1-integrin has been shown to be internalized by the CLIC pathway that is regulated by Cav1 and this paper extends that to argue that this pathway is regulated in response to mechanical stress. It is however not clear how Cav1 regulation of integrin recycling relates to caveolae buffering in response to mechanical stress.

1. While the magnetic bead approach provides a very nice approach to studying local b1-integrin internalization, studies of b1-integrin internalization are for the most part whole cell fixed cell analyses. Some TIRF videos are provided but are not very convincing and are not quantified. Extension of the magnetic bead approach to show local b1-integrin dynamics would be more convincing.

2. There are a number of issues with confocal microscopy. Non-permeabilized paraformaldehyde-fixed cells are presumed to report on surface labeling of the 9GE7 antibody specific for activated integrin. It is not always clear from the text which images were permeabilized or not and it should be noted that paraformaldehyde alone can disrupt the plasma membrane and enable antibody penetration to label cytoplasmic antigen, which often appears as bright localized puncta. The very bright labeling in the hypoosmotic shock treated CAV1 KO cells in figure 7 is suspect; also 1/20 hypoosmotic shock has a larger effect on activated integrin than 1/10 hypoosmotic shock? Some images seem to indicate an ER labeling including the nuclear membrane (i.e. Figure 6D). The only sure way to limit antibody labeling to the surface exposed antigens is to label viable cells at 4C. Colocalization in images that present very diffuse labeling on one channel and a small number of puncta in another is quantified using Pearson's colocalization and used to report on endocytosis. Small changes in normalized colocalization are reported questioning the extent of the reported effect. More appropriate fluorescent-based endocytosis assays are available and should be employed.

*Reviewer #2 (Recommendations for the authors):*

1. Many of the fluorescence images are difficult for the reader to interpret as presented. This is a critical issue given that many of the conclusions of this work rely on quantitative imaging. For example, it is very difficult to assess from overlayed images shown in many panels what structures actually co-localize. A general suggestion would be to provide in the main figures both the merged image and individual images of the two channels being examined, preferably in grayscale, so that the reader can better see the structures in question. Correspondingly, the graphs could easily be made much smaller than shown here.

2. Overall, the quality of the images is not as high as one would expect for an *eLife* paper. For example, in some panels, the fluorescence signal appears to be saturated. This is the case for example in Supplementary Figure 1 F and multiple panels in Supplementary Figure 5. In others, large blotchy structures are present, such as in Figure 7.

3. Much of the quantification of imaging data is normalized, and how it is normalized varies from panel to panel. While in principle there is nothing wrong with this, it makes it difficult for the reader to get a sense of the absolute magnitudes of quantities being measured, such as the degree of colocalization of various markers. This makes it difficult to get an overall sense of flux through various pathways.

4. As a reader I would like to have a better sense of how the numbers add up here to explain the phenotype reported in the first figure.

*Reviewer #3. (Recommendations for the authors):*

A major claim of the manuscript is that the threshold for PM-tension-driven β-1 integrin activation is lower in Cav1 KO cells. However, the experiments as presented are more simply explained by an inability of Cav1 KO cells to buffer membrane tension (already shown by Sinha et al. 2011 Cell), and it is not clear if membrane tension is altered as a consequence of Cav1 KO. This could be addressed by tether pulling experiments or the use of membrane tension probes (e.g. FlipperTR).

In addition, knockdown of talin 1 and 2 is broadly understood to decrease integrin activity across a variety of adherent cell types, and the claim that talin is required for recycling to support active integrin at the cell surface is therefore not well founded.

[Editors’ note: further revisions were suggested prior to acceptance, as described below.]

Thank you for resubmitting your work entitled "Caveolae couple mechanical stress to integrin recycling and activation" for further consideration by *eLife*. Your revised article has been evaluated under the supervision of Suzanne Pfeffer (Senior Editor).

You will be pleased to learn that one reviewer is now fully satisfied and the second believes that the manuscript can be published in *eLife* if you are able to address the following minor issues:

1. The Blot in Figure 1M is still of very low quality and the authors have still not included any compartment markers to determine that their plasma membrane fraction is free of contamination from other compartments. The authors' argument that β 1 integrin can be used as a marker of the plasma membrane obviously falls down in the light of the fact that the authors are studying its trafficking through endosomes and, in doing so, show that an appreciable fraction of this integrin is NOT present at the plasma membrane. Also, there are still no molecular weight markers on this blot or any of the other blots shown in the manuscript. On another note, the blot in Figure 1M is interesting because, if the molecular weight markers run where I think they should, it appears to show that re-expression of Cav1 increases the quantity of mature (120kDa) beta1 integrin at the plasma membrane whereas, in the Cav1KO cells, it is primarily the immature (100kDa) pro-form of beta1 that is at the plasma membrane.

2. There are no scale bars on any of the micrographs

3. I think that reference 46, not 26, is the one for the ELISA-based endocytosis/recycling protocol. Also, I don't think that reference 47 is correct for supporting that beta3 integrin follows a Rab4-dependent "short" loop pathway.

4. The authors have now included, as requested, data showing that re-expression of Cav1 slows-down recycling in Cav1KO cells. But why have the authors shown only active beta1? Surely, they should be showing the recycling data for both total and active beta1 – Indeed, they must have these data as, presumably, they would have split their lysate into ELISA plates coated with antibodies recognising total b1 as well as active beta1. Also, the levels of recycling demonstrated for Cav1KO cells in supplementary Figure 2I differs quite markedly from that shown in Figure 4I.

---

## [Author Response]

[Editors’ note: the authors resubmitted a revised version of the paper for consideration. What follows is the authors’ response to the first round of review.]

Reviewer #1:Caveolae are now widely recognized for their roles in the mechanobiology of the plasma membrane of cells submitted to mechanical stress. In the current manuscript, the authors have studied the crosstalk between caveolae and another mechanosensitive group of proteins, the integrins. Using as a model system mouse embryonic fibroblasts (MEFs) that are deleted for the caveolin-1 gene, it is described that cell adhesion to fibronectin-coated surfaces is increased in the absence of caveolae. The underlying reason for this is identified in the increased presence of the active conformation of beta1 integrin at the cell surface, which results from 2 phenomena: increased endocytic recycling, and increased activation upon mechanical stress.The overall area of research on mechanobiology in relation with membrane trafficking that is addressed in the current study is of high interest to a general readership in the life sciences. The study is based on a rich arsenal of relevant techniques, and represents a substantial amount of work. Some techniques such as cell adhesion stiffness measurements based on magnetic bead oscillation or traction force microscopy are really quite elegant. The manuscript is well written, and pleasant to read.Possibly the most important limitation of the current study is that it remains descriptive on the molecular mechanism aspects. It is clear that the absence of Cav1/caveolae has a number of effects that are clearly defined in the current study. The molecular wiring that underlies these effects remains unexplored at this stage. A few examples: In the absence of Cav1/caveolae, endocytic recycling of beta1 integrin passes from Rab11-dependent slow recycling to Rab4-dependent fast recycling – by which molecular mechanism does Cav1/caveolae connect to the endosomal machinery? Mechanical stress increases beta1 activation in Cav1KO MEFs – by which molecular mechanism does Cav1/caveolae limit this activation in wild-type conditions? Talin is important in the context of this mechanical stress-controlled activation reaction – by which molecular mechanism does Cav1/caveolae interact with talin? Having said this, I still believe that the manuscript will contribute in a substantial manner to the dynamic field of membrane biology research. The following points should be clarified, though.Lines 262-269: The faster beta1 integrin recycling phenotype in Cav1KO MEFs is seen only after 10 min of endocytosis, and not when recycling is measured after 5 min of endocytosis. The authors suggest that "beta1-integrin is stabilized in the presence of Cav1 after 10 minutes endocytosis". How would such stabilization work? The authors most likely don't have the experimental response to this question at this stage. However, it would be helpful if some ideas could be discussed on the molecular mechanisms by which this would work.

We thank the reviewer for these comments. Although we still do not have a complete account as to how Cav1 could affect β 1 integrin stabilization, we have now gathered some evidence showing both increased surface and endosomal (EEA-1 positive) active β 1 integrin in Cav1WT MEFs loaded with cholesterol either by LDL or U18666A treatment, phenocopying Cav1KO phenotype (Suppl. Figure 2J-M). This could be indicative of a Cav1-dependent cholesterol threshold above which beta1 integrin trafficking would be altered. Consistently, previous work in our lab has demonstrated increased exocytic activity upon cholesterol loading in Cav1WT MEFs.

(1). We have now included these ideas both in Suppl. Figure 2 and in the Discussion section.

Lines 315-317: In the paragraphs that precede these lines, the authors present experiments based on the small molecule Cdc42 inhibitor ML141 that lead to the conclusion that in Cav1KO MEFs, part of endocytic uptake occurs by the CLIC/GEEC pathway, while this does not appear to be the case in MEFs that express Cav1. It is concluded that in "wild-type" conditions, part of uptake occurs through caveolae, and that this Cav1-dependent internalization would be compensated by the CLIC/GEEC pathway in the absence of Cav1/caveolae. Can other interpretations be excluded? For example, wouldn't one observe the same phenotype if Cav1/caveolae were to inhibit uptake of a fraction of the beta1 integrin molecules that once liberated in the Cav1KO condition would now be internalized by the CLIC/GEEC pathway? The most direct way to address this point would be electron microscopy to provide ultrastuctural images of the uptake structures in which beta1 integrin is found in the different experimental conditions. To the least, alternative interpretations of the data should be discussed.

We have now included a set of new experiments in Suppl. Figure 3, delineating the relative contribution of different endocytic pathways comparing wild type and Cav1KO MEFs. We now show that active β 1 integrin is mainly endocytosed by Cav1-dependent mechanisms in wild type MEFs as it: (i) co-localized with Cav1 and LacCer, (ii) was significantly reduced upon genistein treatment (a caveolar endocytosis inhibitor (2)) and (iii) was unaffected by ML141 treatment (the CLIC inhibitor), On the other hand, it mainly follows the CLIC-dependent endocytosis in Cav1KO MEFs as it: (i) co-localized with CD44 (a CLIC endocytic marker (5)) and (ii) was significantly reduced upon ML141 treatment as compared to wild type MEFs. Finally, no significant differences were observed in clathrin-dependent β 1 integrin endocytosis between wild type and Cav1KO MEFs. These results might indicate as suggested by the reviewer, that Cav1 is limiting the CLICdependent endocytosis of a pool of active β 1 integrin in wild type MEFs, which becomes available in Cav1KO MEFs. We have now included these ideas in the Discussion section.

Reviewer #2:This manuscript describes the role of Caveolin-1 (Cav1) in integrin recycling using Cav1 knockout MEFs and links integrin endocytosis and recycling to cell mechanosensing and adhesion. I find the subject interesting and, in light of the role of Cav1 and caveolae in the cellular response to mechanical stress, timely and relevant. That being said I find the manuscript to be a compilation of interesting results that lack mechanistic connection between them. We are provided with interesting data on: (1) the role of Cav1 in integrin mechanosensing using an elegant magnetic tweezer approach; (2) Cav1 regulation of integrin endocytosis and recycling but with no direct link to mechanosensing or stretch; (3) the role of talin in regulating CAV1-independent integrin activation. It seems as though Cav1 is altering integrin dynamics and response to mechanical stretch which is quite interesting. While the schematic in figure 8 highlights the Cav1-dependent changes reported in the paper, mechanistic connections between them need to be clarified and I have many questions which I outline below.1. "Cav1/caveolae" Cav1 and caveolae are not the same thing and it cannot be assumed that effects observed in Cav1 KO cells are necessarily attributed to loss of caveolae. I am concerned about use of this term throughout the paper (even the title) and suggest that it would be important to define the specific role of caveolae in the processes described using cell lines expressing Cav1 but not PTRF. Indeed, the only data linking caveolae flattening to the effects shown is Figure 7 and is interesting in that it suggests that lack of membrane buffering by caveolae induces an integrin response. However, to definitively show that this is caveolae dependent and not Cav1 dependent it would be important to use cells expressing Cav1 but not caveolae.

We thank the reviewer for these comments. We have now performed a series of new experiments included in Suppl. Figure 5K-5M where we show that β 1 integrin is still activated in PTRFKO MEFs (that lack caveolae but express some levels of Cav1) after treatment with hypoosmotic shock. These results indicate that lacking the membrane buffering provided by caveolae leads to β 1 integrin activation in Cav1KO MEFs. Accordingly, we have now changed Cav1/caveolae for caveolae throughout the text, including the title.

2. The studies on b-integrin endocytosis switch to a CLIC pathway in Cav1 KO is interesting and supports a role for Cav1 as an inhibitor of CLIC endocytosis. It would be important to provide more evidence for CLIC endocytosis than inhibition of Cdc42. Is b-integrin cointernalized with CD44? Is this pathway CD44- or raft-dependent? Can the effects of Cav1 on integrin mechanosensing be attributed to CLIC endocytosis of integrin? How much of total surface integrin is internalized via this pathway and not clathrin or caveolin pathways? Is CLIC (and fluid phase) endocytosis generally upregulated in CAV1 KO cells? If the endocytosis effects are most clearly seen at early times of cell spreading how does this relate to the mechanosensing experiments done on spread cells?

We have now included a series of new co-localization studies in Suppl. Figure 3 delineating the relative contribution of different endocytic routes to β 1 integrin endocytosis. Specifically, we now show that anti-active β 1 integrin antibody colocalizes with Cav1 after 3 minutes of endocytosis at 37ºC, and this entry is significantly reduced by genistein treatment (a common caveolar endocytosis inhibitor (2)) but it is unaffected by ML141 treatment (CLIC inhibitor). Likewise, BODIPY-LacCer (a caveolar endocytic marker (3, 4)) co-localizes with endogenous active β 1 integrin (9EG7) after 3 minutes of endocytosis at 37ºC, which is also significantly reduced by genistein treatment. On the other hand, anti-active β 1 integrin antibody colocalizes with anti-CD44 antibody (a CLIC endocytic marker (5)) after 3 minutes of endocytosis at 37ºC, and this entry is significantly blocked by ML141 in Cav1KO MEFs but not in wild type MEFs. Finally, we observed no significant differences in the co-localization of anti-active β 1 integrin and transferrin (a clathrin endocytic marker) after 3 minutes of endocytosis at 37ºC in wild type and Cav1KO MEFs. Altogether, these results suggest that CLIC-dependent β 1 integrin endocytosis is blocked in wild type MEFs, where β 1 integrin is mainly endocytosed by caveolae, and becomes free in Cav1KO MEFs, where it is mainly endocytosed by CLICs.

Regarding cell spreading: as stated in material and methods section where we described the endocytosis/recycling assay, “cells were incubated in complete DMEM for 2 hours at 37^o^C”, this was the shortest time-point require for cells to stand assay conditions, and also matches the magnetic tweezers measurements experimental settings, where cells were allowed to spread for a short period of time (30 minutes). We have now included this information in the corresponding material and method section.

3. I am also a little confused as to how the authors envisage Cav1 regulating integrin recycling. Is Cav co-internalized with b-integrin and stabilizes it in endosomes, slowing recycling? Is this occurring in caveosomes? If this is the case it would be important to show it. Can the TIRF videos be quantified to support the ELISA data?

We still do not know the exact mechanism by which Cav1 is regulating integrin recycling, however we suspect it depends on Cav1-dependent cholesterol organization as we have obtained some results showing both increased surface and endosomal (EEA-1 positive) active-β 1 integrin upon loading Cav1WT MEFs with cholesterol either by LDL or U18666A treatment, phenocopying Cav1KO phenotype (Suppl. Figure 2J-M). This might indicate that Cav1 determines a cholesterol threshold above which beta1 integrin trafficking is altered. Accordingly, previous work in our lab has demonstrated increased exocytic activity upon cholesterol loading in Cav1WT MEFs (1), which in our conditions would affect β integrin trafficking dynamics. We have now included these ideas in Suppl. Figure 2J-M and in the Discussion section.

Regarding TIRF videos, we have now included new experiments and quantifications (Suppl. Figure 2C-F and Suppl. Videos 6 and Video 7) showing that integrin trafficking in Cav1WT MEFs resembles that observed in Cav1KO MEFs after treatment with high hypoosmotic pressure conditions. These results might indicate that increasing plasma membrane tension beyond caveolae buffering capacity changes integrin trafficking dynamics.

4. If the issues related to Cav1 vs caveolae mentioned in 1 are addressed, caveolae membrane buffering may be limiting integrin activation and an integrin-dependent tensive response. This is very interesting. Caveolae membrane buffering is thought to be an early response to mechanical stress so shouldn't WT cells show the same effect at longer times or with increased hypoosomotic pressure? Can this be tested? And what about other mechanical stressors?

We have now included a set of new experiments pertaining caveolae buffering and integrin activation after hypoosmotic treatment in suppl. Figure 5N-5W, where we show that integrin activation also occurs in Cav1WT MEFs after both longer treatment times and with increased hypoosmotic pressure, as compared to Cav1KO MEFs. These results suggest that caveolae membrane buffering is limiting integrin activation before a plasma membrane tension threshold is reached.

5. Finally, we are shown in Figure 7 that Talin regulates adhesion and b-integrin activation in KO cells. Is Talin required for the increased integrin dynamics shown in previous figures?

To answer this question, we have performed two complementary experiments now included in Supp. Figure 6F-6M: (1) co-localization studies of 9EG7 (for testing endogenous active β 1 integrin) or anti-active beta1-488 antibody (for testing integrin endocytosis) and EEA-1 (early endosomal marker), and (2) biotinylation assays of surface-active β 1 integrin in Talin-1 and 2silenced Cav1KO MEFs. Interestingly, surface active β 1 integrin is reduced and consistently increased intracellularly in EEA-1 positive endosomes, upon talin1/2 knockdown. Collectively, these new results suggest that talin is modulating not only adhesion and integrin activation but also integrin trafficking in Cav1KO MEFs.

Reviewer #3:This paper from Lolo et al. described experimentation aimed at determining how caveolin and caveolae influence integrin function. The use mouse embryonic fibroblasts (MEFs), and caveolin null MEFs that have, or have not been rescued with exogenously expressed caveolin. They first use magnetic tweezers to show that knockout of caveolin influences integrin mechanical behaviour. They then used a surface biotinylation assay to look at the endocytosis and endocytic recycling of beta1 integrin. They found that caveolin null MEFs had unaffected levels of integrin endocytosis, but that they recycled bet1 integrin more rapidly than wild-type MEFs. They also show that a small pool of beta1 integrin is internalised through a cdc42-dependent pathway and attribute this to the clathrin independent carrier (CLIC) pathway, but it is unclear from these data whether this is influenced by caveolin. The authors proceed from this to use immunofluorescence colocalisation studies and dominant negative mutants of Rab GTPases to determine that caveolin null MEFs may have altered integrin trafficking, possibly via a Rab4-dependent route. Finally, the authors show that hypoosmotic shock (to put the plasma membrane under tension) leads to increased integrin activation in caveolin null MEFs and that this may be talin-dependent.In general this paper describes a series of observations which are not sufficiently well-connected to support the synthesis that the authors assemble for how caveolin may regulate integrin trafficking and function. Moreover, much of the experimentation and the way that the results are presented are neither convincing nor of the standard necessary for publication in eLife.1. Figure 4. The authors have used a surface biotinylation method to show that integrin recycling is different in wild-type and caveolin null MEFs. These assays need to be performed with the rescue cells too.

We thank the reviewer for all these comments. We have now included a set of new recycling experiments in Supp. Figure 2I, comparing Cav1KO and Cav1KO+Cav1 reconstituted MEFs. The results now show that recycling is slow down upon Cav1 reconstitution indicating that the phenotype is Cav1 specific.

2. The authors pursue their recycling results using immunofluorescence (Figure 6) to try to infer that differences in recycling, shown in Figure 4 are Rab4-dependent. I do not find any of the data in Figure 6 convincing. It is not clear to me why the various colocalisations tested represent recycling of integrins. Moreover, how can one look at colocalisation with dominant negative mutants of Rabs (which are known to localise aberrantly) and conclude that a particular cargo is in any given compartment?

We thank the reviewer for pointing out the mistake with 9EG7 and Rab11 co-localization under Rab11 DN transfection. We have now repeated the experiment and performed a co-localization between 9EG7 and LBPA (lysobisphosphatidic acid, a late endosomal marker) instead of Rab11. We now show that blocking Rab11-dependent recycling affect β 1 trafficking both in wild type and Cav1KO MEFs, whereas blocking Rab4-dependent recycling only affect β 1 trafficking in Cav1KO MEFs. Regarding the quality of proof of co-localization studies, it is important to stress that Figure 4 (co-localization studies to analyze endosomal β 1 integrin) has to be read in conjunction with Suppl. Figure 4 (surface biotinylation assays to analyze changes in surface β 1 integrin), as referred to in the text. The former (Figure 4) shows how blocking two well-known recycling pathways (“long-loop”, Rab11-dependent and “short-loop”, Rab4-dependent) leads to a differential endosomal accumulation of active β 1 integrin in wild type and Cav1KO MEFs; the latter (Suppl. Figure 4), on the other hand, reveals a differential reduction in active β 1 integrin surface levels under the same experimental conditions. Combining both results led us to propose the most likely conclusion, i.e. that β 1 integrin preferentially follows a Rab11-dependent recycling pathway in wild type MEFs, whereas it is partially switched to a fast, Rab4-dependent recycling in Cav1KO MEFs.

3. How are the data in Figure 5 consistent with those in Figure 4? The authors show in Figure 4 that integrin endocytosis is not affected by caveolin knockout. In Figure 5, they show that an inhibitor of the CLIC pathway (although they do not validate its efficacy or specificity) slightly inhibits integrin endocytosis. They also show that fluid-phase endocytosis is increased in in caveolin null MEFs and infer that integrins are internalised through this pathway. But, in Figure 4 integrin endocytosis is not different between caveolin wild-type and null fibroblasts. Thus, I am completely at a loss to interpret the authors' concluding statement which reads: 'Thus, in the absence of Cav1/caveolae, early endocytosis of b1-integrin occurs at least in part through fluid-phase uptake, which provides an alternative entry route that would compensate for lack of Cav1-dependent internalization.'

We have now clarified this point by including a series of new co-localization studies in Suppl. Figure 3, delineating the relative contribution of different endocytic routes to β 1 integrin endocytosis. Specifically, we now show that anti-active β 1 integrin antibody colocalizes with Cav1 after 3 minutes of endocytosis at 37ºC, and this entry is significantly reduced by genistein treatment (a common caveolar endocytosis inhibitor (2)) but it is unaffected by ML141 treatment (CLIC inhibitor). Likewise, BODIPY-LacCer (a caveolar endocytic marker (3, 4)) co-localizes with endogenous active β 1 integrin (9EG7) after 3 minutes of endocytosis at 37ºC, which is also significantly reduced by genistein treatment. On the other hand, anti-active β 1 integrin antibody colocalizes with anti-CD44 antibody (a CLIC endocytic marker (5)) after 3 minutes of endocytosis at 37ºC, and this entry is significantly blocked by ML141 in Cav1KO MEFs (which proves ML141 specificity, blocking CLIC endocytosis) but not in wild type MEFs. Finally, we observed no significant differences in the co-localization of anti-active β 1 integrin and transferrin (a clathrin endocytic marker) after 3 minutes of endocytosis at 37ºC in wild type and Cav1KO MEFs. Altogether, these results suggest that CLIC-dependent β 1 integrin endocytosis is blocked by Cav1 in wild type MEFs, where it is mainly endocytosed by caveolae, and becomes free in Cav1KO MEFs, where it is mainly endocytosed by CLICs.

4. There are no attempts, as far as I can see, to link the alterations in integrin trafficking to their activation. I would have thought that, having shown that integrin trafficking, their mechanical properties and their activation-state (9EG7 binding) are different between caveolin null and wild-type cells, the authors would want to attempt to determine which of these events are mechanistically connected. At least, for eLife I would expect some of these questions to be addressed.

We have now included new experiments aiming at mechanistically connect integrin activation and trafficking dynamics:

1. We have performed new TIRF experiments and quantifications (Suppl. Figure 2C-F) showing that integrin trafficking in Cav1WT MEFs resembles that observed in Cav1KO MEFs after treatment with high hypoosmotic pressure conditions. These results might indicate that increasing plasma membrane tension beyond caveolae buffering capacity changes integrin trafficking dynamics.

2. Regarding talin activity, we have performed two complementary experiments now included in Supp. Figure 6F-6M: (1) co-localization studies of 9EG7 (for testing endogenous active β 1 integrin) or anti-active beta1-488 antibody (for testing integrin endocytosis) and EEA-1 (early endosomal marker), and (2) biotinylation assays of surface-active β 1 integrin in Talin-1 and 2silenced Cav1KO MEFs. Interestingly, surface active β 1 integrin is reduced and consistently increased intracellularly in EEA-1 positive endosomes, upon talin1/2 knockdown. Collectively, these new results suggest that talin is modulating not only adhesion and integrin activation but also integrin trafficking in Cav1KO MEFs.

5. The 9EG7 staining data presented in Figure 7 are really not convinding. I do not see how the authors can quantify images such as those shown in Figure 7D. The staining just looks like a splodge of fluorescence implying some kind of aggregation, or gross disruption of the cell membrane. Also, some of this staining looks to be at the cell surface (Figure 7L), and some is clearly intracellular (Figure 7I), so how can this necessarily reflect active, talin-bound integrin that is competent to bind ligand?

All the immunofluorescences shown in Figure 7 were done following the extracellular staining described in material and methods, i.e., after fixation, cells were immediately incubated with the primary antibody without permeabilization, so the majority of the signal comes from surface active β 1 integrin (9EG7 antibody). We show the difference between permeabilized and nonpermeabilized samples in Supp. Figure 1 (please compare 1H and 1I with 1H and 1I). However, to definitively show that β 1 integrin is still active and capable of ligand binding, we have now included new immunostaining of talin and soluble Fibronectin in Cav1KO MEFs after hypoosmotic treatment. Suppl. Figure 6B-6E shows co-localization between β 1 integrin (9EG7 antibody) and both fibronectin-FITC (6B and 6C) and talin (6D and 6E) which proves its active conformation.

6. The presentation of much of the data in this manuscript is not of the standard that is necessary for acceptance in eLife. Many of the axes are not labelled (such as the y-axes in Figure 4F – I), or there are labels/designations that are unclear; for instance, what does 'recycling by timepoint 0 after 10 min endocytosis' mean (Figure 4G)? Also, there are metrics which are not described at all. For instance, I cannot seem to find in the manuscript (or in reference 19) what 'reinforcement increment' is and how it is obtained. Is it the same as 'relative stiffening' in reference 19?

We apologize for these shortcomings. We have now included axes labels where necessary and better explained how recycling is represented. Reinforcement increment refers to the relative change in reinforcement, calculated as the difference between the last and initial measurements; we have now clarified this point in figure legend.

7. The hugely increased 'reinforcement increment' of ConA beads in Figure 2H over fibronectin beads in Figure 2G doesn't seem to square with the authors' assertion that integrins couple the beads to the cytoskeleton to restrict their movement. It looks to me that these data indicate that the ConA beads are much more tightly associated with the cytoskeletal machinery than the fibronectin-coated ones.

Concanavalin A is a lectin that binds sugars, as those presented in glycosylated proteins and lipids within the cellular surface; therefore, conA-coated beads provide a non-specific binding as compared to FN-coated beads, which specifically bind to integrins. Assuming that the average density of surface sugars is higher than integrins, ConA-coated beads surface adhesion will be higher than FN-coated ones, resulting in higher absolute reinforcements.

8. To support their conclusion that Rho is not involved in Cav1 knockout mediated reinforcement (Figure 3C) don't the authors need to show that the reinforcement increment still increases when one compares Cav1+/+;p190KD and Cav1-/-; p190KD cells? Curiously, the authors have not shown this.

The same Cav1KO MEFs used in this manuscript have been extensively studied in our laboratory (10-12). We have previously showed that Cav1 modulates cell contraction by regulating specifically Rho activity through p190RhoGAP localization to the PM. In the absence of Cav1, PM-localized p190RhoGAP increases, leading to a significant reduction in Rho activity, dampening cellular contractility (as we show by traction force microscopy, Figure 3D) and related processes. Accordingly, knocking down p190RhoGAP in Cav1KO MEFs completely rescues all phenotypes we have evaluated so far (10, 12). The fact that we observed no significant differences in magnetic tweezers measurements as compared to traction force (Figure 3C and 3D), speaks of a minor role of Rho in early cellular reinforcement. Consistently, different reports have previously shown initial integrin adhesion in the absence of cytoskeleton connection, suggesting that early mechanosensing could be locally triggered (13-15). We have now included these ideas and references in the discussion.

References

I. N.-L. Lucas Albacete-Albacete, Juan Antonio López , Inés Martín-Padura, Alma M. Astudillo, Alessia Ferrarini, Michael Van-Der-Heyden, Jesús Balsinde , Gertraud Orend , Jesús Vázquez , Miguel Ángel del Pozo, ECM deposition is driven by caveolin-1–dependent regulation of exosomal biogenesis and cargo sorting. Journal of cell biology 11, (2020).

[Editors’ note: what follows is the authors’ response to the second round of review.]

Comments to the Authors:We realize that you went to great effort to try to improve a prior manuscript but sadly must write to say that, after consultation with the reviewers, we have decided that this manuscript will not be considered further for publication by Life.Reviewer #1 (Recommendations for the authors):This paper proposes to link caveolin-1 regulation of the response to mechanical stress to beta1-integrin recycling. While some of the data are interesting, in particular, the Cav1-dependent response of cells to magnetic beads, other aspects of the paper are less than convincing. B1-integrin has been shown to be internalized by the CLIC pathway that is regulated by Cav1 and this paper extends that to argue that this pathway is regulated in response to mechanical stress. It is however not clear how Cav1 regulation of integrin recycling relates to caveolae buffering in response to mechanical stress.

We thank the reviewer for her/his positive appreciation of the magnetic tweezers studies. We have now performed new experiments to specifically connect integrin recycling with the buffering ability of caveolae (new panels in Figure 6P-6V).

1. While the magnetic bead approach provides a very nice approach to studying local b1-integrin internalization, studies of b1-integrin internalization are for the most part whole cell fixed cell analyses. Some TIRF videos are provided but are not very convincing and are not quantified. Extension of the magnetic bead approach to show local b1-integrin dynamics would be more convincing.

We thank the reviewer for these comments. Although an important part of the beta1-integrin internalization studies is based on whole cell stainings after fixation, we have also performed an extensive array of in vivo experiments, examples can be found in: (i) Figure 4D-4I and Suppl. Figure 2G2I, with ELISA-based endocytosis/recycling assays, and (ii) Suppl. Figure 2J, Suppl. Figure 4A-4C and Suppl. Figure 6V, where surface active beta1-integrin is analyzed after altering trafficking dynamics by changing cholesterol homeostasis (Suppl. Figure 2J), or different components of the recycling machinery (Suppl. Figure 4A-4C and Suppl. Figure 6V).

We sincerely apologize if TIRF-related information was not clearly conveyed, as we consider it an important addition to support our main conclusions. We have indeed provided quantifications of TIRF videos, as shown in Suppl. Figure 2C-2F and explained in Supplementary information: “Quantification of TIRF videos show normalized fluorescence integrated density (IntDen) over frames (Suppl. Figure 2C-2E). Graph represents the mean of the difference between normalized fluorescence integrated density of adjacent frames (frame_x_-frame_x-1_) (Suppl. Figure 2F). IntDen was calculated as in the following formula: IntDen= Raw Integrated density (sum of pixel values in selection) x (Area in scaled units)/ (Area in pixels), which was then normalized by the IntDen mean of all frames analyzed”. We have now clarified this in the main text. We also provide two additional TIRF videos comparing Cav1WT MEFs transfected with beta1-GFP before and after hypoosmotic treatment (with DMEM diluted 1:20 in distilled water). We hope differences across conditions are clearer now.

Finally, as an extension of the magnetic bead experiments shown in Figure 2G and 2H, we also performed some magnetic twisting experiments (as shown in Suppl. Figure 5A-5J), which also allow for probing cell micromechanics. Results showed a significant tension-induced increase in beta1-integrin activation in Cav1KO MEFs as compared to Cav1WT MEFs, supporting results from other experiments and reinforcing the main conclusion of the manuscript, i.e., that caveolae regulate both integrin activation and recycling. However, to further support our claims, we have now performed a new set of experiments to specifically link integrin recycling to caveolae buffering in response to mechanical stress. To do so, we incubated Cav1KO and Cav1WT MEFs with anti-active beta1-Alexa 488 antibody for 1 hour at 4^o^C followed by 3 minutes endocytosis at 37^o^C. Immediately after that, we challenged cells for 1 minute with either DMEM diluted 1:10 (which can be buffered by caveolae flattening, as previously described, Sinha et al. Cell 2011, and is shown in our own results in both Figure 7A-7F and Suppl. Figure 5N-5W), or DMEM diluted 1:20 in distilled water, which exceeds the buffering capacity of caveolae (as we have shown in Suppl. Figure 5N-5W). To remove any remaining surface fluorescence, two quick steps of acid stripping were done prior to fixation with PFA. Strikingly, whereas Cav1KO MEFs showed a significant reduction in the endocytosed active beta1-integrin pool both at 1:10 and 1:20 dilutions, Cav1WT MEFs only showed a similar significant reduction at 1:20 dilution (new panels in Figure 6P-6V). These results indicate that caveolae buffering prevents integrin recycling below a certain force threshold, and therefore regulate integrin dynamics in response to mechanical stress.

2. There are a number of issues with confocal microscopy. Non-permeabilized paraformaldehyde-fixed cells are presumed to report on surface labeling of the 9GE7 antibody specific for activated integrin. It is not always clear from the text which images were permeabilized or not and it should be noted that paraformaldehyde alone can disrupt the plasma membrane and enable antibody penetration to label cytoplasmic antigen, which often appears as bright localized puncta. The very bright labeling in the hypoosmotic shock treated CAV1 KO cells in figure 7 is suspect; also 1/20 hypoosmotic shock has a larger effect on activated integrin than 1/10 hypoosmotic shock? Some images seem to indicate an ER labeling including the nuclear membrane (i.e. Figure 6D). The only sure way to limit antibody labeling to the surface exposed antigens is to label viable cells at 4C. Colocalization in images that present very diffuse labeling on one channel and a small number of puncta in another is quantified using Pearson's colocalization and used to report on endocytosis. Small changes in normalized colocalization are reported questioning the extent of the reported effect. More appropriate fluorescent-based endocytosis assays are available and should be employed.

Reviewer 1 raises an important point regarding surface staining, for which we are grateful. Following this recommendation we have repeated the analysis by incubating cells with 9EG7 antibody at 4^o^C for 1h. Although some of the previous cytoplasmic integrin staining was lost, results are analogous to those previously observed (new panels in Suppl. Figure 6A-6J). Additionally, we have clarified in figures legends whether extracellular staining was done before or after fixation. According to previous observations in the lab (unpublished data, currently under review in another journal), the extension in osmolarity reduction correlates with plasma membrane tension increase. This implies that 1/20 dilution leads to higher plasma membrane tension increase than 1/10 dilution, and therefore could induce a larger effect on integrin activation, which is what we have observed. Altogether, these results suggest that caveolae membrane buffering is limiting integrin activation before a plasma membrane tension threshold is reached.

As rightly indicated by the referee, CLIC-dependent beta1-integrin endocytosis has previously been connected to Cav1 regulation, and our co-localizations studies were meant to confirm these results. However, to better show the co-localizing structures, we now provide both the merged image and the individual images of the two channels in all figures. Additionally, to avoid possible bias of normalized Pearson’s correlations, we have quantified mean fluorescence intensity of endocytosed beta1-integrin (followed by anti-active beta1-Alexa 488 antibody, as now shown in Suppl. Figure 3D´ and 3L´) after treatment with either genistein (a Cav1-dependent endocytosis inhibitor) or ML^-1^41 (a CLIC-dependent endocytosis inhibitor). In line with our previous observations, beta1-integrin endocytosis was significantly reduced in Cav1WT MEFs after treatment with genistein, whereas it was unaffected after treatment with ML^-1^41. In contrast, beta1-integrin endocytosis was significantly reduced in Cav1KO MEFs treated with the CLIC inhibitor, ML^-1^41. Again, these results indicate that beta1-integrin endocytosis is mainly Cav1-dependent in Cav1WT MEFs and mainly CLIC-dependent in Cav1KO MEFs. We have now included these new analyses in Suppl. Figure 3D´and 3L´.

Reviewer #2 (Recommendations for the authors):1. Many of the fluorescence images are difficult for the reader to interpret as presented. This is a critical issue given that many of the conclusions of this work rely on quantitative imaging. For example, it is very difficult to assess from overlayed images shown in many panels what structures actually co-localize. A general suggestion would be to provide in the main figures both the merged image and individual images of the two channels being examined, preferably in grayscale, so that the reader can better see the structures in question. Correspondingly, the graphs could easily be made much smaller than shown here.

We thank the reviewer for these comments. We have now modified all figures to show both merged as well as separated channels to better see how the different structures co-localize.

2. Overall, the quality of the images is not as high as one would expect for an eLife paper. For example, in some panels, the fluorescence signal appears to be saturated. This is the case for example in Supplementary Figure 1 F and multiple panels in Supplementary Figure 5. In others, large blotchy structures are present, such as in Figure 7.

We apologize for this, as embedding figures throughout the main text affected their overall resolution. We have now removed them and included as separated files to improve their quality. It is true, as fairly indicated by the reviewer, that some images show fluorescence saturation and the presence of large blotchy structures within some integrin stainings; however, in order to compare signals across conditions we were forced to use the same laser settings, that together with the large dynamic range of intensities within the integrin signal, led to such unavoidable problems. We have now repeated the analysis by incubating cells with 9EG7 antibody at 4^o^C for 1h prior to fixation, which reduced any potential antibody penetration and cytoplasmic signal accumulation. Of note, resulting images, now free of any aggregated structures, led to analogous results as those previously observed (new panels in Suppl. Figure 6A-6J). Additionally, we have also improved the quality of diagrams in Figures 8 and Suppl. Figure 6W.

3. Much of the quantification of imaging data is normalized, and how it is normalized varies from panel to panel. While in principle there is nothing wrong with this, it makes it difficult for the reader to get a sense of the absolute magnitudes of quantities being measured, such as the degree of colocalization of various markers. This makes it difficult to get an overall sense of flux through various pathways.

We thank the reviewer for these comments and apologize for having made such confusion. We have now better clarified both in figures and captions the corresponding normalization procedure used. We hope it is now easier to get the overall flow of results. Generally, in co-localization studies we have normalized each Pearson’s correlation coefficient by the mean of the control condition. In those experiments where we quantified fluorescence intensity, values were normalized to mean fluorescence intensity of the control condition when analyzing anti-active beta1-Alexa 488 antibody signal, or to each analyzed area and finally referred to area control when analyzing 9EG7 antibody signal (for more details, please have a look at accompanying raw data excel files).

4. As a reader I would like to have a better sense of how the numbers add up here to explain the phenotype reported in the first figure.

We are sorry but figure 1 does not contain any quantification, we please ask the reviewer if she/he can better specify the figure he/she is referring to.

[Editors’ note: what follows is the authors’ response to the third round of review.]

You will be pleased to learn that one reviewer is now fully satisfied and the second believes that the manuscript can be published in eLife if you are able to address the following minor issues:1. The Blot in Figure 1M is still of very low quality and the authors have still not included any compartment markers to determine that their plasma membrane fraction is free of contamination from other compartments. The authors' argument that β 1 integrin can be used as a marker of the plasma membrane obviously falls down in the light of the fact that the authors are studying its trafficking through endosomes and, in doing so, show that an appreciable fraction of this integrin is NOT present at the plasma membrane. Also, there are still no molecular weight markers on this blot or any of the other blots shown in the manuscript. On another note, the blot in Figure 1M is interesting because, if the molecular weight markers run where I think they should, it appears to show that re-expression of Cav1 increases the quantity of mature (120kDa) beta1 integrin at the plasma membrane whereas, in the Cav1KO cells, it is primarily the immature (100kDa) pro-form of beta1 that is at the plasma membrane.

We apologize for the low quality of the blot shown in Figure 1M. We have now repeated plasma membrane purification, substituting β 1 integrin with Transferrin Receptor as a plasma membrane (PM) marker (now in revised Figure 1M). We have now included molecular markers in all blots, additionally, whole unprocessed gels are included as source data for visual inspection. We thank the reviewer for his/her interesting observation on integrin maturation, as it is quite certain that Cav1 re-expression in Cav1KO MEFs seems to increase mature β 1 integrin (molecular markers were indeed running where the reviewer was thinking). Although we have now replaced beta1 integrin with Transferrin Receptor as a PM marker, as aforementioned, the difference in maturation levels may well correlate, according to previous reports^1^, with the random-like migration we have previously reported for Cav1KO MEFs^2^, but this will be the scope of future studies.

2. There are no scale bars on any of the micrographs

We apologize for having omitted this information. We have now included scale bars when appropriate in figure micrographs, with the corresponding indications in figure legends.

3. I think that reference 46, not 26, is the one for the ELISA-based endocytosis/recycling protocol. Also, I don't think that reference 47 is correct for supporting that beta3 integrin follows a Rab4-dependent "short" loop pathway.

We thank the reviewer for these comments. We have followed the capture ELISA for determination of biotinylated integrins as indicated in reference 26, but it is true that a similar protocol was previously described by Prof. Jim Norman in reference 46, we have then included both references. We have eliminated reference 47 when referring to beta3 short-loop recycling, leaving only 45 and 46.

4. The authors have now included, as requested, data showing that re-expression of Cav1 slows-down recycling in Cav1KO cells. But why have the authors shown only active beta1? Surely, they should be showing the recycling data for both total and active beta1 – Indeed, they must have these data as, presumably, they would have split their lysate into ELISA plates coated with antibodies recognising total b1 as well as active beta1. Also, the levels of recycling demonstrated for Cav1KO cells in supplementary Figure 2I differs quite markedly from that shown in Figure 4I.

We thank the reviewer for these comments. We have indeed split the lysate but we only analyzed active β 1 integrin as it was the most relevant pool for explaining the increased reinforcement observed in our magnetic tweezers studies. However, we have now performed the corresponding ELISA for total β 1 integrin. To our surprise no significant differences between Cav1KO and Cav1KO+Cav1 MEFs are observed. This might indicate, as it usually happens with rescue experiments, that Cav1 re-expression only restores the wild type situation partially, rescuing the recycling of active β 1 integrin pool, but being unable to restore the inactive one. We have now added these results in Suppl. Figure 2J (now Figure 4—figure supplement 1J). Finally, although we have always performed the recycling assay under the same experimental conditions, there is a certain level of variability among experimental replicates: different cell passages, variations in total surface biotinylation, etc., which might explain the differences in recycling levels between Suppl. Figure 2I (now Figure 4—figure supplement 2J) and Figure 4I indicated by the reviewer.

References

1. Guo, M. et al. Altered processing of integrin receptors during keratinocyte activation. Exp Cell Res 195, 315-322, doi:10.1016/0014-4827(91)90379-9 (1991).

2. Grande-Garcia, A. et al. Caveolin-1 regulates cell polarization and directional migration through Src kinase and Rho GTPases. J Cell Biol 177, 683-694, doi:10.1083/jcb.200701006 (2007).